# A single-cell transcriptional atlas reveals resident progenitor cell niche functions in TMJ disc development and injury

Ruiye Bi[1,5], Qing Yin[1,2,5], Haohan Li[1,5], Xianni Yang[1], Yiru Wang [1], Qianli Li[1], Han Fang[1], Peiran Li[1], Ping Lyu[3], Yi Fan[3], Binbin Ying[4] & Songsong Zhu [1] ✉

The biological characteristics of the temporomandibular joint disc involve complex cellular network in cell identity and extracellular matrix composition to modulate jaw function. The lack of a detailed characterization of the network severely limits the development of targeted therapies for temporomandibular joint-related diseases. Here we profiled single-cell transcriptomes of disc cells from mice at different postnatal stages, finding that the fibroblast population could be divided into chondrogenic and non-chondrogenic clusters. We also find that the resident mural cell population is the source of disc progenitors, characterized by ubiquitously active expression of the NOTCH3 and THY1 pathways. Lineage tracing reveals that *Myh11*[+] mural cells coordinate angiogenesis during disc injury but lost their progenitor characteristics and ultimately become *Sfrp2*[+] non-chondrogenic fibroblasts instead of *Chad*[+] chondrogenic fibroblasts. Overall, we reveal multiple insights into the coordinated development of disc cells and are the first to describe the resident mural cell progenitor during disc injury.

The temporomandibular joint (TMJ) is a unique joint in the craniofacial system that is essential for normal masticatory functions, including talking, chewing, swallowing and facial expression. The TMJ is mainly composed of the articular condyle, articular disc, articular fossa, fibrous capsule, and synovial membrane. A crucial element in the TMJ, the articular disc is a dense and versatile fibrocartilaginous structure that facilitates load bearing and congruity and prevents bone-to-bone contact throughout chewing movement[1]. A dysfunctional TMJ disc is the most prevalent TMJ disorder and manifests as disc displacement, thinning and perforation[2]. These pathological conditions of the disc often act as antecedents to a series of degenerative changes that can affect the entire TMJ, ultimately causing symptomatic TMJ disarrangement that affects 5%-30% of people worldwide, with an estimated annual health cost of $4 billion[2–4].

The articular disc has long been known to be an avascular and noninnervated tissue;[1] therefore, its low regeneration capacity has been the greatest challenge for the restoration of TMJ disarrangement, with the current clinical strategy being limited to pain management, followed by eventual joint replacement[5]. In contrast to that of organic diseases in other fibrocartilage tissues, such as the meniscus and intervertebral disc, the complex etiology of TMJ disc damage is still unclear due to limited knowledge of the cell types, functions, and molecular networks within the disc. Multiple studies have investigated the cellular compositions of TMJ discs in different species, including rats[6], rabbits[7], pigs[8], primates[6,9] and humans[10,11]. These studies simply divided cells into fibroblasts (FBs) and chondrocyte-like cells, whose cell proportions and matrix secretion behavior change with aging and in the context of osteoarthritis (OA). However, disc cells have distinct

[1]State Key Laboratory of Oral Diseases, National Clinical Research Center for Oral Diseases, Department of Orthognathic and TMJ Surgery, West China Hospital of Stomatology, Sichuan University, Chengdu 610041, China. [2]Max-Planck Institute for Heart and Lung Research, W. G. Kerckhoff Institute, Bad Nauheim D-61231, Germany. [3]State Key Laboratory of Oral Diseases, National Clinical Research Center for Oral Diseases, Department of Operative Dentistry and Endodontics, West China Hospital of Stomatology, Sichuan University, Chengdu 610041, China. [4]Department of Stomatology, Ningbo First Hospital, 59 Liuting street, Ningbo 315000, China. [5]These authors contributed equally: Ruiye Bi, Qing Yin, Haohan Li. ✉e-mail: ZSS_1977@163.com

spatial specializations, with a collagen fiber network oriented in different directions, reflecting varying mechanical properties. These dynamic temporal-spatial changes in TMJ discs under physiological and pathological conditions can clearly not be fully explained by the current rough cell type classification. The lack of a detailed characterization of the complex cellular network in the disc severely limits the development of innovative targeted therapies and biotherapies for TMJ-related diseases.

Recently, many works have investigated different fibrocartilage and hyaline cartilage tissues by using single-cell analyses[12–16]. Understanding these tissues at the single-cell level can provide insights into the onset and progression of pathology. Considering our limited understanding of the cellular composition of TMJ discs over the past decade, defining disc subpopulations at the single-cell level would both deepen our understanding of the developmental process of TMJ discs and improve the likelihood of generating pluripotent cell derivatives for biotherapy and tissue engineering for this delicate tissue.

Here, we performed a comprehensive single-cell RNA sequencing (scRNA-seq) analysis of mouse TMJ disc tissues across development. We identified four principal cell types and 9 clusters with diverse transcript expression characteristics and cellular functions. Among the cell clusters, the cells in the most clusters, FBs, are heterogeneous in terms of biological processes related to ECM secretion and metabolism. The cells in the endothelial cell (EC) cluster and macrophage (MPh) cluster are mainly distributed in the junction of the anterior band and attachment, suggesting potential roles of this region in the inflammatory response. More importantly, a distinct mural cell (MC) cluster was found to have the characteristics of mesenchymal pluripotent cells for TMJ disc injury repair. In summary, this work provides a critical resource for understanding the cellular components and signaling networks defining TMJ disc development and can be used to aid in the molecular identification of different cell subtypes. This work will enable the development of new approaches to TMJ disc restoration and support the use of the most regenerative cells from TMJ discs in cell-based therapies.

## Results

### scRNA-Seq captures four major TMJ disc cell populations

To dissect the cellular composition and transcriptional changes of the murine TMJ disc during postnatal development, we obtained mouse TMJ disc tissues at different postnatal developmental stages (newborn stage: 3-day-old (d3); juvenile stage: 3-week-old (3 w); adult stage: 16-week-old (16 w); aged stage: 78-week-old (78 w)) and profiled the disc cells by scRNA-seq. We prepared single-cell suspensions from TMJ disc tissues and performed scRNA-seq using the 10x Genomics platform (Fig. 1a).

A total of 43481 cells were captured, of which 39111 cells were retained after quality control analysis. In total, we captured the expression of 21340 genes. Dimensionality reduction approaches were used to visualize all 39111 cells in low-dimensional space, in which each cell's coordinates were estimated to preserve the expression similarity in UMAP plots. After unsupervised graph clustering of the 4 datasets combined (3 d, 3 w, 16 w, and 78 w), the Seurat 3 R-Package was used to segregate the captured cells into 9 distinct cell clusters, which were classified into four principal cell types according to the cell assemblies on the UMAP plot: the fibroblast (FB) cluster, the macrophage (MPh) cluster, the endothelial cell (EC) cluster, and mural cell (MC) cluster (Fig. 1b, c). We plotted the fraction of cells expressing each marker across all stages using the genes in the heatmap of the top 20 differentially expressed genes (DEGs) of the four principal cell types[13,17–23] (Fig. 1d, and S1a). We also provide examples of the many additional markers that were used to define these cell types in Fig. S1b–h.

Previous reports showed that the cellular composition of different tissues and cell types tends to vary with age[24]. In our TMJ disc samples, we investigated how the fractions of different cell clusters changes during postnatal growth and aging. We found that the four principal cell types had relatively conserved cell proportions, and all principal cell types were found to be widely expressed from newborn (3 d) to aged (78 w) mice (Fig. 1e, f). We also found that the numbers of the 4 cell types at different stages and the numbers of a single cell type at different stages were proportionate (Fig. 1g, h). This finding signified that the 4 types of cells constantly maintained their functions over the postnatal life cycle of mice. On the other hand, the transcript expression patterns of the cell clusters were dynamic. The DEGs of the four principal cell types had marked expression variations (Fig. S1a–h). For example, the average expression levels of several FB DEGs, such as *Prg4* (5.5:1) and *Cilp* (9.1:1), were several times higher in aged (78 w) mice than in newborn (3 d) mice. At the same time, the levels of some FB DEGs, such as *Ptn* (1:0.51) and *Col1a1* (1:0.48), were significantly higher in 3-day-old mice than in 78-week-old mice (Fig. S1c). This phenomenon was also observed for multiple DEGs of the other principal cell types (Fig. S1e, g, i). This observation showed that a specific type of cell may play distinct roles at different developmental stages via fine transcriptional regulation in a single cell.

### FBs show heterogeneity in TMJ disc development and aging

When dissecting the TMJ discs from mice at different postnatal stages, the diameters of the discs were strikingly increased (by 3.0 times from 3 d to 16 w), in contrast with observations for many other tissues with a relatively limited size increase from the neonatal stage, such as the brain and most visceral organs (Fig. 2a, b). Since the regulation of organ size is based on progenitor cell numbers and growth factors that determine cell diversity to meet the need for functionality in vertebrates[25], this phenomenon signified that the large increase in disc size during postnatal development is a general representation of active proliferation and matrix secretion of disc cells, probably as a result of adaptation of jaw functions for chewing and phonating after birth. To further investigate the mechanism underlying cellular activity related to disc function and morphology at the single-cell level, we focused on FBs, which are considered the main cell component in TMJ discs in different species[6,8,9,11]. Our scRNA-Seq data showed that the FB clusters consisted of 88.0% of the total cells (34422/39111, Fig. 2c). Unsupervised clustering of FBs identified 6 distinct subclusters. To determine the identity of the FB tissues included in our samples, we utilized several recent transcriptomic studies of murine and human FBs[17–19] and gene ontology (GO) analyses as our guide (Fig. S1a, and S2a, b). We found multiple biological processes and pathways that were differentially regulated during postnatal development and metabolism in TMJ discs. Notably, GO analyses showed that the enriched biological processes of multiple FB clusters are closely related to ECM organization and functions (Fig. S2b). Thus, we investigated the expression patterns of ECM markers in FBs. Within the FB fraction, different clusters exhibited various ECM signatures; for example, DEGs of the FB1 cluster included *Eln*, *Postn*, *Serpinf1* and *Col3a1*, DEGs of the FB4 cluster included *Col22a1* and *Fn1*, and DEGs of the FB6 cluster included *Chad* and *Cilp2* (Fig. S2a). Meanwhile, the heatmap of ECM expression showed a dynamic expression distribution in different FB clusters (Fig. 2d). For example, in FB1 and FB2 clusters, *Col5a1*, *Col12a1*, *Col14a1* and *Fndc1* were highly expressed, which are genes associated with fibrillogenesis and fibrillary homeostasis in connective tissues[26–29]. In the FB5 and FB6 clusters, *Chad*, *Comp* and *Cilp* appeared to be more active ECM markers, which are genes associated with chondrogenesis and injury repair in fibrocartilage[30,31]. *Co22a1* is especially active in the FB4 cluster, suggesting potential functions related to myotendinous junctions and vascular stability in this cell population[32–34] (Fig. 2d).

We also compared ECM features of TMJ discs at different postnatal stages (Fig. S3a). At 3 d, *Col12a1*, *Col14a1* and *Dpt* were highly expressed. At 78 w, *Fn1*, *Comp* and *Cilp* were actively expressed. Among the different stages, discs at 3 w were found to have the most abundant and active ECM expression. These discs showed high

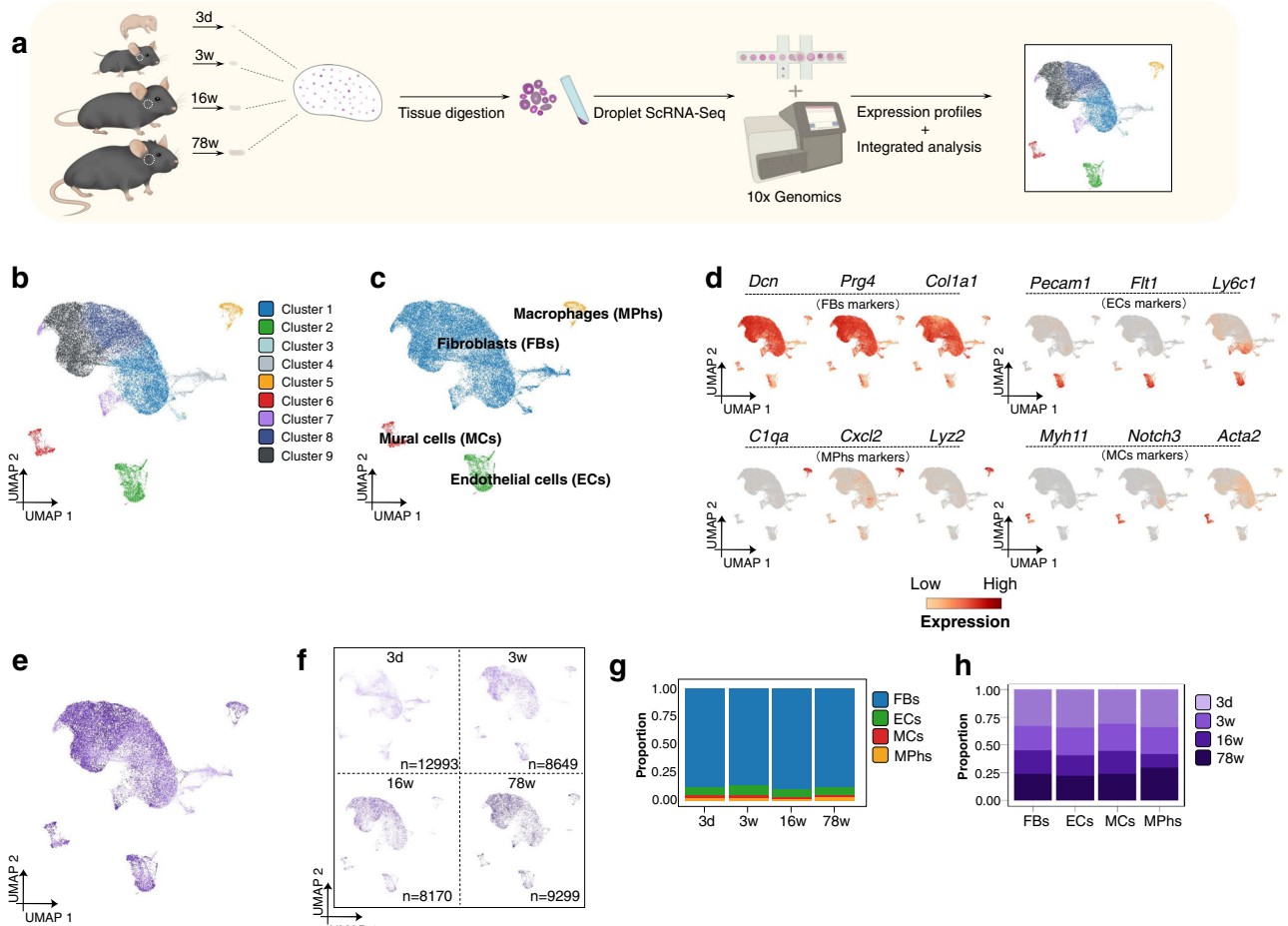

**Fig. 1 | scRNA-Seq captures four main cell populations in mouse TMJ disc.**
**a** Schematic of the mouse TMJ disc tissues dissected for single-cell transcriptomic analyses. **b** Dimension reduction presentation (via UMAP) of combined single-cell transcriptome data from TMJ discs of newborn, juvenile, adult and aged mice (*n* = 39111). Each dot represents a single cell and is labeled with corresponding cell categories and colored according to its cell type identity. Clusters were generated using a resolution of 0.2 prior to subclustering into major cell types according to the Methods. The Seurat 3 R-Package segregation grouped the cells into 9 distinct cell clusters. **c** Disc cells were classified into four principal cell types according to cell assemblies on the UMAP plot. Blue: fibroblasts (FBs), *n* = 34420; yellow: macrophages (MPhs), *n* = 2780; green: endothelial cells (ECs), *n* = 1030; and red: mural cells (MCs), *n* = 881. **d** Expression patterns of selected markers projected on the

UMAP plot (**a**). For each main cell type, cell type-defining genes identified by manual review are shown to represent the location of the cluster in UMAP. **e** Dimension reduction presentation (via UMAP) of integrated single-cell transcriptome data for all stages. Each dot represents a single cell and is labeled by different color depths according to stage. **f** Dimension reduction presentation (via UMAP) of single-cell transcriptome data at each stage (3 d, 3 w, 16 w, and 78 w). Each dot represents a single cell and is labeled by different color depths according to stage. **g** Bar graph of relative cell proportions by mouse stage. Blue: fibroblasts (FBs); yellow: macrophages (MPhs); green: endothelial cells (ECs); and red: mural cells (MCs). **h** Bar graph of the relative cell proportions of the main cell types, showing the contribution of each tissue stage to each cell type. Cell proportions were labeled by different color depths according to stage.

expression levels of both the early-stage markers *Col3a1* and *Col5a1* and the late-stage markers *Comp* and *Dcn*, which are vital for the structural stability of the ECM and mechanical strength sensing of fibrocartilage[35–38]. In other words, discs at this stage, while growing in size via cell proliferation, also become sturdier in order to respond to mechanical force by secreting specific ECM components. This may be because C57 mice have a change in food from breast milk to solid food at 2–3 weeks[39].

Next, to validate our scRNA-seq profiles of ECMs at the protein level, we performed Movat pentachrome staining of cross-sections of TMJ tissues. The results showed varying trends of different ECM components within discs at different postnatal stages. Collagen fibers were highly expressed at 3 w, proteoglycan was abundant at the newborn stage (3 d), and the myofiber/fibrin area was increased at the adult (16 w) and aged (78 w) stages (Fig. S3b, c). Consistent with the collagen fiber changes observed by Movat pentachrome staining, immunohistochemical staining of collagen I, which is encoded by a gene (*Col1a1*) that is a common shared DEG among FBs, also showed that the 3-week-old mice had the most abundant collagen I expression

(Fig. S3d). This ECM expression heterogeneity among different stages and FB clusters highlights the adaptive differentiation of FBs during TMJ function and physiological degeneration[6].

Previous studies divided TMJ disc cells into fibroblasts and chondrocyte-like cells. However, there was no separated cell type defined as chondrocyte cluster in the UMAP. To locate the chondrocyte-like cells in our single cell data, we used previous published marker genes of chondrogenic differentiation or cartilage formation to identify the chondrocyte-like cells in our 4 main cell types[40–42]. We found that expressions of both chondrogenic differentiation markers such as *Sox5*, *Sox6*, *Sox9* and chondrocyte metabolism markers such as *Chad*, *Comp*, *Col11a1* were enriched in FBs cluster (Fig. 2e). When we further compared expressions of these chondrocyte related markers between FB subclusters, we found that expressions of these genes were mainly enriched in FB5 and FB6 (Fig. 2f). Guided by GO analyses, FB5 and FB6 were also found closely related to chondrocyte differentiation and ossification (Fig. S2b). Therefore, we defined these 2 FB clusters as chondrogenesis related fibroblast 1 and 2, and other 4 FB clusters as non-chondrogenic fibroblasts, with additional definition according to

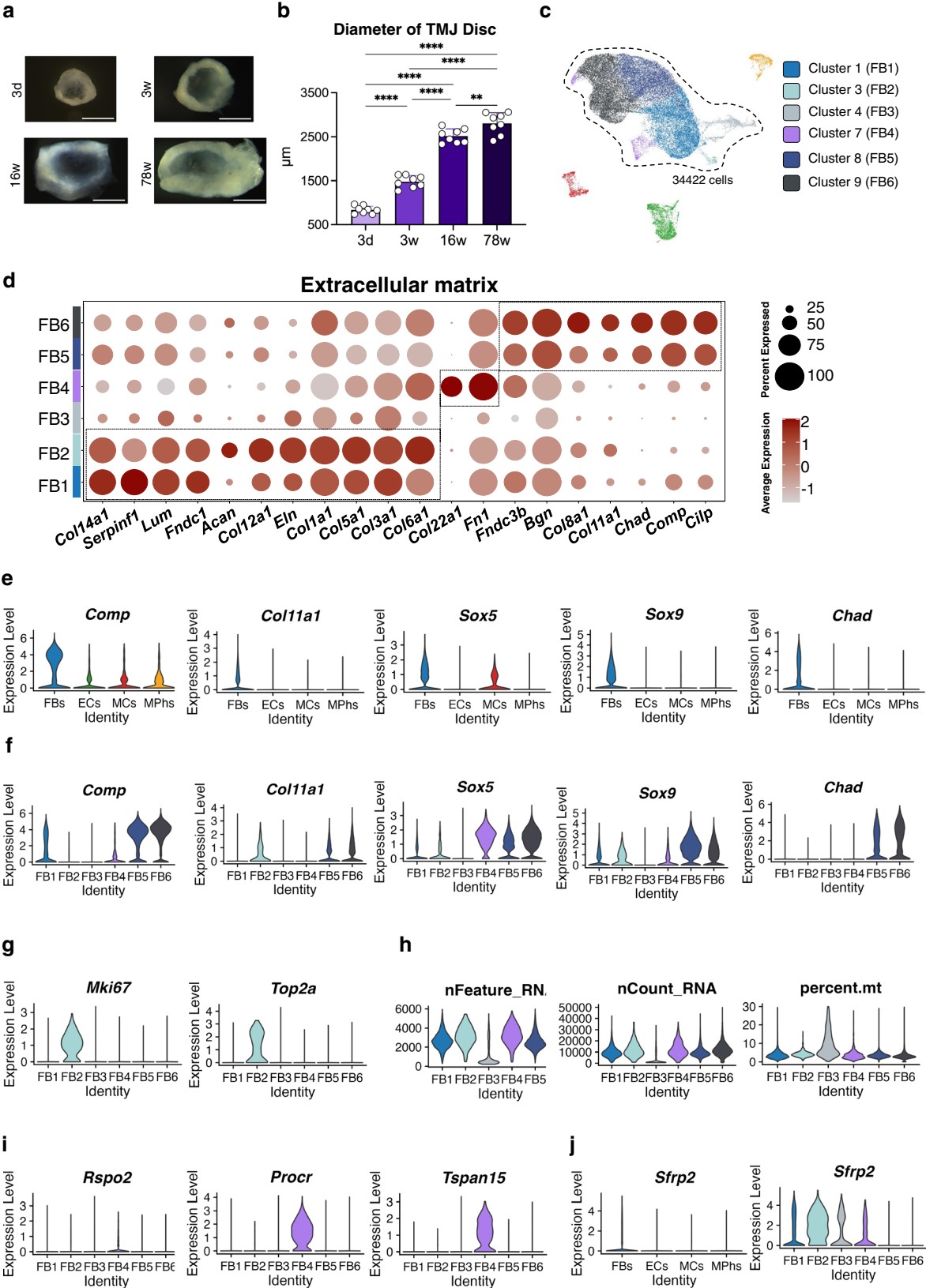

their DEGs. For instance, FB2 was defined as proliferation related non-chondrogenic FB cluster for its high expression of *Mki67* and *Top2a*[43,44], FB3 was defined as apoptosis related non-chondrogenic FB cluster for increased percentage of mitochondria[45], and FB4 was defined as chondrogenesis inhibition related FB cluster for specific expression of *Tspan15*, *Procr* and *Rspo2*[42] (Fig. 2g–i).

To verify our findings by bioinformatic analyses, we singled out *Chad* as the DEG of chondrogenesis related FBs, and *Sfrp2* as the DEG of non-chondrogenic FBs, for further cellular localization of different clusters (Fig. 2f, j). RNA FISH showed that *Sfrp2*⁺/*Chad*⁻ non-chondrogenic fibroblasts were located at the border of anterior and posterior bands of the disc, near the attachment, while *Sfrp2*⁻/*Chad*⁺

**Fig. 2 | The differentially expressed gene pattern distinguishes 6 fibroblast cell clusters. a** TMJ disc tissues were dissected at different stages. Scale bar = 500 μm. $N = 8$ independent animals at each stage with similar results. **b** The diameter of each disc was measured under a stereoscopic microscope. Data are presented as mean values +/-SD. $N = 8$ independent animals at each stage. The one-way ANOVA with Tukey's multiple comparison test was used, $^{**}p = 0.0097$, $^{****}p < 0.0001$. Scale bar: 500 μm. **c** UMAP of fibroblast clusters (FB1-FB6). Clusters were generated using a resolution of 0.2 prior to subclustering into major cell types according to the Methods. **d** Extracellular matrix markers of FBs. Blue: fibroblasts (FBs); yellow: macrophages (MPhs); green: endothelial cells (ECs); and red: mural cells (MCs). **e** Violin plot of the expression of selected differentially expressed genes (DEGs) of

chondrogenesis related FB clusters in the four principal cell types. Expression values are nonnormalized. **f** Violin plot of the expression of selected DEGs of chondrogenesis related FB clusters in different FB clusters. Expression values are nonnormalized. **g** Violin plot of the expression of selected DEGs of non-chondrogenic FB cluster (proliferation). Expression values are nonnormalized. **h** Violin plot of the expression of selected DEGs of non-chondrogenic FB cluster (apoptosis). Expression values are nonnormalized. **i** Violin plot of the expression of selected DEGs of non-chondrogenic FB cluster (chondrogenesis inhibition). Expression values are nonnormalized. **j** Violin plot of the expression of *Sfrp2*. Expression values are nonnormalized.

chondrocyte-like disc cells were mainly located in the intermediate zone, which finding was consistent with previous reports by transmission electron microscopy[8]. At the same time, we found that the proportion of *Sfrp2*⁺ fibroblasts in the articular disc was gradually decreased during postnatal growth and aging, while the proportion of *Chad*⁺ fibroblasts showed opposite trends (Fig. S4a–c).

To further clarify diverse roles of FB clusters at different postnatal stages, we chose our 3d scRNA-seq data, which is the most active stage for cell development and differentiation, and used CytoTRACE (v.0.3.3) to predict differentiation states of different FB clusters at this stage[46]. We found that the non-chondrogenic fibroblast 2 (proliferation) cluster showed the highest CytoTRACE value, which was estimated with optimum developmental potential[46]. On the other hand, we also found the chondrogenesis related FB1 and FB2 were at the relatively later order of CytoTRACE value (Fig. S5a–f). These findings enlighten us again that the chondrocyte-like FBs in TMJ disc are terminally differentiated cells with limited capacity of self-repair and being associated with aging and injury[6,10].

### Spatial-temporal expressions of MPhs and ECs in the TMJ disc

Among the four principal cell types, the MPh cluster and EC cluster were both ubiquitously present from the newborn stage to the aged stage (Fig. 1e–h). The MPh cluster exhibited classic MPh markers, including the complement-related genes *C1qa*, *C1qb*, and *C1qc*[21], the chemokine-related genes *Ccl2*, *Ccl3*, *Ccl4*, *Ccl7*, and *Cxcl2*[47], and the classic M1/M2 cytokines *Tnf* and *Mrc1*[20] (Fig. S1a). Guided by these findings from single-cell analysis, we investigated the potential spatial and temporal distribution trends of MPHs (Fig. 3a). We singled out C1QA as the staining marker for MPhs, as its coding gene (*C1qa*) is one of the most specific DEGs for the MPh cluster (Figs. S1a and 3b). Immunofluorescence staining showed that C1QA⁺ MPhs were mainly distributed in the synovium of the joint, and the C1QA⁺ cells in the disc were mainly distributed in the junction of the anterior band/anterior attachment and posterior band/posterior attachment (bilaminar region) at all stages. In the medial band, few C1QA⁺ cells were seen, especially at earlier stages (Fig. 3c). At the same time, the C1QA⁺ cell proportions observed by FCM gradually decreased from 3 d (9.3 ± 0.3%) until later stages (3 w: 8.8 ± 0.5%, 16 w: 8.0 ± 0.6%, and 78-82 w: 6.8 ± 0.5%) (Fig. 3d).

DEGs of the EC cluster include the typical endothelial markers *Fabp4*, *Pecam1*, *Ly6c1*, *Ly6e*, etc[22]. To investigate the distribution trends of ECs, we used PECAM-1, which is encoded by *Pecam1*, as a staining marker for labeling ECs in TMJ discs and observed the distribution features of ECs in the discs (Fig. 3e, f). Immunofluorescence staining showed that there were few PECAM-1⁺ cells in the intermediate zone of the disc body, which is consistent with a previous report that the TMJ disc is an avascular organ[1]. However, there were still some PECAM-1⁺ ECs clustering in the junction area of the anterior band/anterior attachment. During postnatal growth and aging, there was a trend of gradually decreasing PECAM-1⁺ EC expression (Fig. 3g). Consistently, FCM of PECAM-1⁺ cells from aged mice (78-82 w) revealed a smaller EC proportion that of the other stages (Fig. 3h), which is in accordance with the decreased abundance of C1QA⁺ monocytes in these mice.

Since angiogenesis has critical roles in progenitor fate changes and tissue degeneration in fibrocartilage[48,49], we can conclude that the variation in PECAM-1⁺ EC proportions at different stages could be relevant to disc development and repair.

### MC subclusters have two pseudotime lineage directions

Among the four principal cell types of TMJ discs, MCs were found to highly express multiple mesenchymal stem cell markers, such as *Myh11*, *Mylk*, and *Acta2*[13,50–52] (Fig. S1a). To further analyze the commonalities and characteristics of mesenchymal cell types in this cluster that might promote and support TMJ disc development and injury repair, we re-clustered the MC population and obtained 6 subclusters (Fig. 4a, b). A heatmap of DEGs and GO analyses for each cluster allowed us to assign probable identities to each cell type (Fig. 4b, Fig. S6a, b). The expression heatmap of multipotent differentiation genes in MC subclusters and subsequent pseudotime analysis revealed that the MC1 cluster, situated at the earliest pseudotime, was endowed with the strongest multipotent differentiation capacity (Fig. 4c) and later transcriptionally bifurcates into two distinct lineage fates (Fig. 4d, f). Notably, the early MC cluster (MC1) was composed of cells mostly from 3 d and then comparably from 3 w, 16 w and 78 w (Fig. S6c). Examination of known markers suggested that 2 distinct developmental paths exist in this trajectory: the lineage fate of multipotent maintenance of stem cell characteristics (*Myh11*, *Mylk*, *Acta2*, *Myl9*) and the fate of adaptive differentiation toward functioning disc cells (*Comp*, *Prg4*, *Sox9*, *Cxcl14*) (Fig. 4e, g, h, Fig. S6d). Thus, our computational analysis suggests that MCs are a heterogeneous cluster with distinct pseudotime lineage fates involving either self-renewal or functional differentiation.

### NOTCH/THY1 is activated in MCs in disc development and aging

We aimed to dissect how cellular interactions within the TMJ disc affect the regulation of signaling pathways detected in TMJ discs by employing cell–cell interaction analyses. We screened signaling pathways that are potentially involved in cell interactions between different disc clusters. Multiple pathways, such as NOTCH, VTN, THY1, IL6, ANGPTL and GRN, were screened out by specific activation in cell communications between MCs and other clusters (Fig. 5a, b). Among these pathways, THY1 and NOTCH were invariably activated at all postnatal stages, suggesting that these two molecular pathways are vital for both disc cell growth and homeostasis (Fig. 5a, b). Regarding interactions between clusters, most THY1 ligand–receptor interactions were seen between MCs, MPhs and several specific FB clusters (FB1 and FB4) (Fig. 5c). However, NOTCH signaling, which is particularly active via both ligands and receptors in MCs, was more complicated. NOTCH signaling interactions were observed between different clusters, while self-interaction within the MC cluster was found to be the strongest NOTCH signaling interaction (Fig. 5d). Meanwhile, self-interaction of NOTCH was extremely frequent at earlier stages (3 d, 3 w) and gradually decreased in frequency with aging (Fig. 5d). The dynamic NOTCH pathway interactions in TMJ discs highlighted their role in physiological and pathological processes of synovial joints[53].

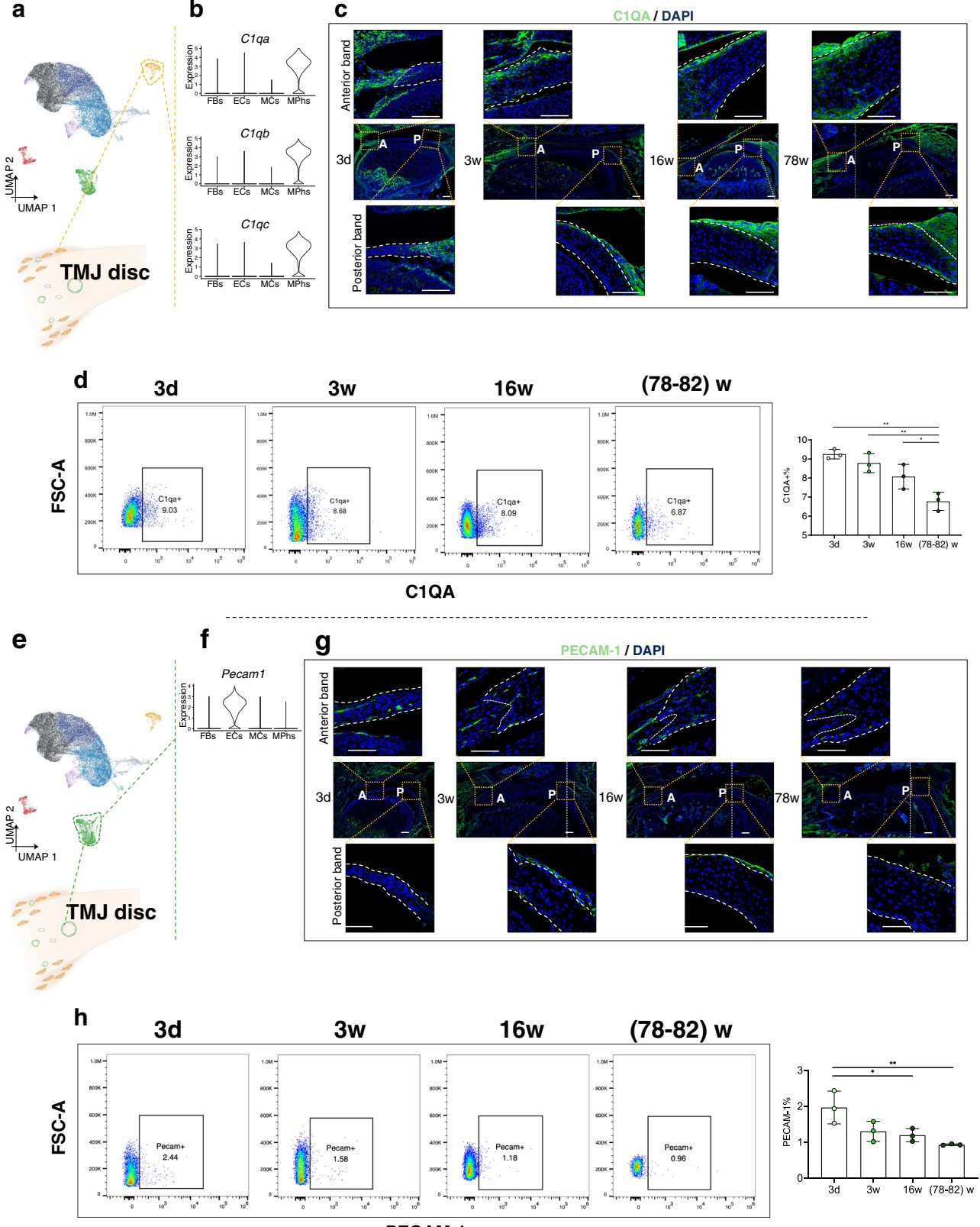

## THY1+/NOTCH3+ MCs show skeletal progenitor capacities

A feature plot of *Notch* genes in UMAP showed that *Notch3* was the most important DEG for MCs among the *Notch* family members (Fig. 6a). *Notch3* expression remained relatively stable at all stages, while *Thy1* expression gradually decreased with aging in mice (Fig. 6b). Immunohistochemical staining showed that NOTCH3+/THY1+ cells

were mainly distributed in the middle of the anterior band, close to the anterior attachment. The expression of both NOTCH3 and THY1 gradually decreased with aging; however, THY1 expression was decreased in a much more drastic manner, and THY1+ cells could barely be observed in the disc area in aged mice (Fig. S7a, b). To confirm this phenomenon, FCM was performed using disc cells at different stages

**Fig. 3 | Spatial-temporal expressions of the macrophage cluster and endothelial cluster in the TMJ disc. a** UMAP plot of 9 clusters (refer to Methods for the codes used). The macrophage cluster (MPhs, Cluster 2) is marked in yellow. **b** Violin plot of *C1qa/C1qb/C1qc* in the 4 principal cell types. Expression values are non-normalized. **c** Representative images of TMJ discs subjected to immuno-fluorescence staining for C1QA at different stages. Green: C1QA; blue: DAPI. A: anterior band, P: posterior band, white dotted lines: boundary of the discs. $N = 6$ independent animals with similar results. Scale bar: 50 μm. **d** Flow cytometry plot of MPhs stained with C1QA antibodies. The dot plots display representative data (left). The quantifications of %C1QA[+] cells (mean +/-SD) at different stages are presented (right). $N = 3$ independent biological samples. The one-way ANOVA with Tukey's multiple comparison test was used for analysis, [*]$p = 0.0413$, [**](3d vs. 78-82w)

$p = 0.0012$, [**](3w vs. 78-82w) $p = 0.0044$. Abbreviations: FSC-A: forward scatter area. **e** UMAP plot of 9 clusters (refer to Methods for the codes used). The endothelial cluster (ECs, Cluster 5) is marked in green. **f** Violin plot of *Pecam1* in the 4 principal cell types. Expression values are nonnormalized. **g** Representative images of TMJ discs subjected to immunofluorescence staining for PECAM-1 at different stages. Green: PECAM-1; blue: DAPI. A: anterior band, P: posterior band, white dotted lines: boundary of the discs. $N = 6$ independent animals with similar results. Scale bar: 50 μm. **h** Flow cytometry plot of ECs stained with PECAM-1 antibodies. The dot plots display representative data (left). The quantifications of % PECAM-1[+] cells (mean +/-SD) at different stages are presented (right). $N = 3$ independent biological samples. The one-way ANOVA with Tukey's multiple comparison test was used for data analysis, [*]$p = 0.0414$, [**]$p = 0.0089$. Abbreviations: FSC-A: forward scatter area.

(Fig. 6c). The percentage of THY1[+] cells in TMJ discs was 12.4 ± 2.0% at 3 d and was dramatically decreased to <6% in the adult and aged stages. At the same time, the percentage of NOTCH3[+] cells was comparatively stable at all stages (Fig. 6c). Considering that MCs have progenitor potential during tissue development and injury[23,54], these findings support the hypothesis that mouse TMJ discs gradually lose their progenitor capacity during aging.

To further validate the skeletal progenitor potential of disc MCs and to understand the mechanism underlying the dramatic changes in THY1 expression in TMJ discs, stem cell characteristics were analyzed using THY1[+] MCs in 3-day-old mice. Compared with THY1[-] cells, THY1[+] MCs had stronger skeletal progenitor characteristics, including higher colony formation potential and proliferative capacities (Fig. S8a–c). During tri-lineage transduced differentiation in vitro, THY1[+] MCs were also found to have markedly stronger potential for osteogenesis, adipogenesis and chondrogenesis (Fig. 6d, e, S8d–f). Ex vivo transplantation of freshly magnetic-activated cell sorting (MACS_)-purified THY1[+] MCs from 3 d discs under the renal capsules of NOD-SCID mice was performed to assess heterotopic cell differentiation. Four-week grafts of THY1[+] MC populations formed more collagen (yellow), fibronectin (red) and mucin (blue), however, there was no spontaneous mature chondrocytes formation found in the renal capsule ossicles. As expected, THY1[-] disc cells were not able to proliferate and differentiate to form osteogenic or chondrogenic cell niches in the ex vivo transplants (Fig. 6f).

**MCs transform toward non-chondrogenic FBs in TMJ disc injury**
We next sought to investigate the function of MCs in disc injury and repair. A surgically induced disc injury mouse model was generated and used for preparing single-cell suspensions of injured disc tissues. 7716 cells were retained from injured disc cells after quality control analysis and were integratedly analyzed with normal disc cells (Fig. 7a). The UMAP plot showed that injured disc cells were also composed of 4 principal cell types: FBs, MCs, MPhs and ECs (Fig. 7b, c). However, the inferred signaling interaction numbers were decreased from average 3209 in normal discs to 1203 in injured discs, and the interaction strength was also found decreased in injured disc (1.248 vs. average 2.119 in normal discs) (Fig. 7d, e). We screened signaling pathways that are most potentially involved in cell interactions between different cell clusters in injured TMJ discs (Fig. S9a–d), as well as the specifically increased signaling pathway in MCs after injury model generation (Fig. 7f–j). Both results pointed to NOTCH signaling pathway as the target signaling for MCs expressional and functional changes after disc injury.

Then, to test whether the TMJ disc homeostasis and repair involved tissue-resident MCs, we performed lineage tracing of MCs using fluorescence reporter animals driven by a MC-specific marker, *Myh11* (*Myh11-Cre^{ER}; Tm^{fl/-}* mice, Fig. 8a). To rule out the possibility of a circulating source of disc tissue repair, we used a parabiosis model that generated circulation between GFP[+] mice and TMJ injured non-GFP mice (Fig. S10a). We found there was no circulating GFP[+] cells participating in joint disc repair (Fig. S10b, c). Then we compared RFP[+] MC

lineage distributions and expression features between surgically induced discs and normal TMJ discs (Fig. 8b–g). Under a fluorescence microscope, MC lineage cell numbers were significantly increased in injured discs. For the most part, RFP[+] cells were located in the anterior band in the normal disc but migrated from the junction of the disc band/attachment to the intermediate band, along with PECAM-1[+] vascular proliferation, in the injured disc (Fig. 8d, g). Interestingly, while the MC lineage proliferated and migrated into the middle zone of the injured disc, RFP[+] cells gradually lost their MC characteristics, which manifested as a loss of NOTCH3 expression and THY1 expression (Fig. 8h–k, Supplementary Movie 1–4), as well as a loss of *Myh11* expressions itself in surgical induced mouse disc injury (Fig. S11a, b). Interestingly, we also found that when RFP[+] MCs lineage migrate toward disc injury site, considerable number of RFP[+] MCs lineage started to express *Sfrp2* (RFP[+]/*Sfrp2*[+]), but much fewer RFP[+]/*Chad*[+] cells were observed (Fig. 8l, m). Notably, at the terminus of the newly formed blood vessels in the injured disc band, RFP[+] MC lineage cells were no longer adjacent to PECAM-1[+] ECs, which indicates that these resident 'perivascular cell' lineages ultimately transformed into other cell types functioning in cell metabolism and injury repair.

## Discussion
In this study, we built the first single-cell atlas for TMJ discs at different postnatal stages. Our work details the complex landscape of disc cell types and provides comprehensive insights into cell function in this unique fibrocartilage within the craniofacial system. In this disc atlas, four main cell types and 9 distinct clusters were found, revealing a previously unappreciated diversity of FBs, the most abundant cells, within the disc. We also found that different FB clusters exhibit distinct DEGs, and could be classified as chondrogenesis related FB clusters and non-chondrogenic FB clusters, which classification coordinates previous studies that divided disc cells into chondrocyte-like cells and fibroblasts. The chondrogenesis related FB clusters locate around intermedial zone of TMJ disc with enriched expression of chondrocyte ECM markers. These features, in addition to the latter order of Cyto-TRACE values, suggest these terminally differentiated chondrocyte like cells are functional for mechanical loading bearing and ECM homeostasis with limited capacity of self-repair. On the other hand, the non-chondrogenic FB clusters with higher CytoTRACE values, mainly locate near the anterior and posterior band attachment, and support more diverse DEG features such as cell proliferation and chondrogenesis inhibition. These features suggest non-chondrogenic FBs is the more active cell type in articular disc, potentially with high developmental and repair capacity. When comparing ECM feature changes at different stages, we find the most abundant ECM components are actively expressed In 3 w discs, which probably supports adaptation of mechanical strength for jaw functions when the food source changes from milk to solid food at 2–3 weeks[39]. This heterogeneity provides evidence that physiological demand drives cell heterogeneity in organs at different developmental stages.

The majority of temporomandibular joint dysfunctions (TMJDs) initiate with anterior disc displacement (ADD), during which the

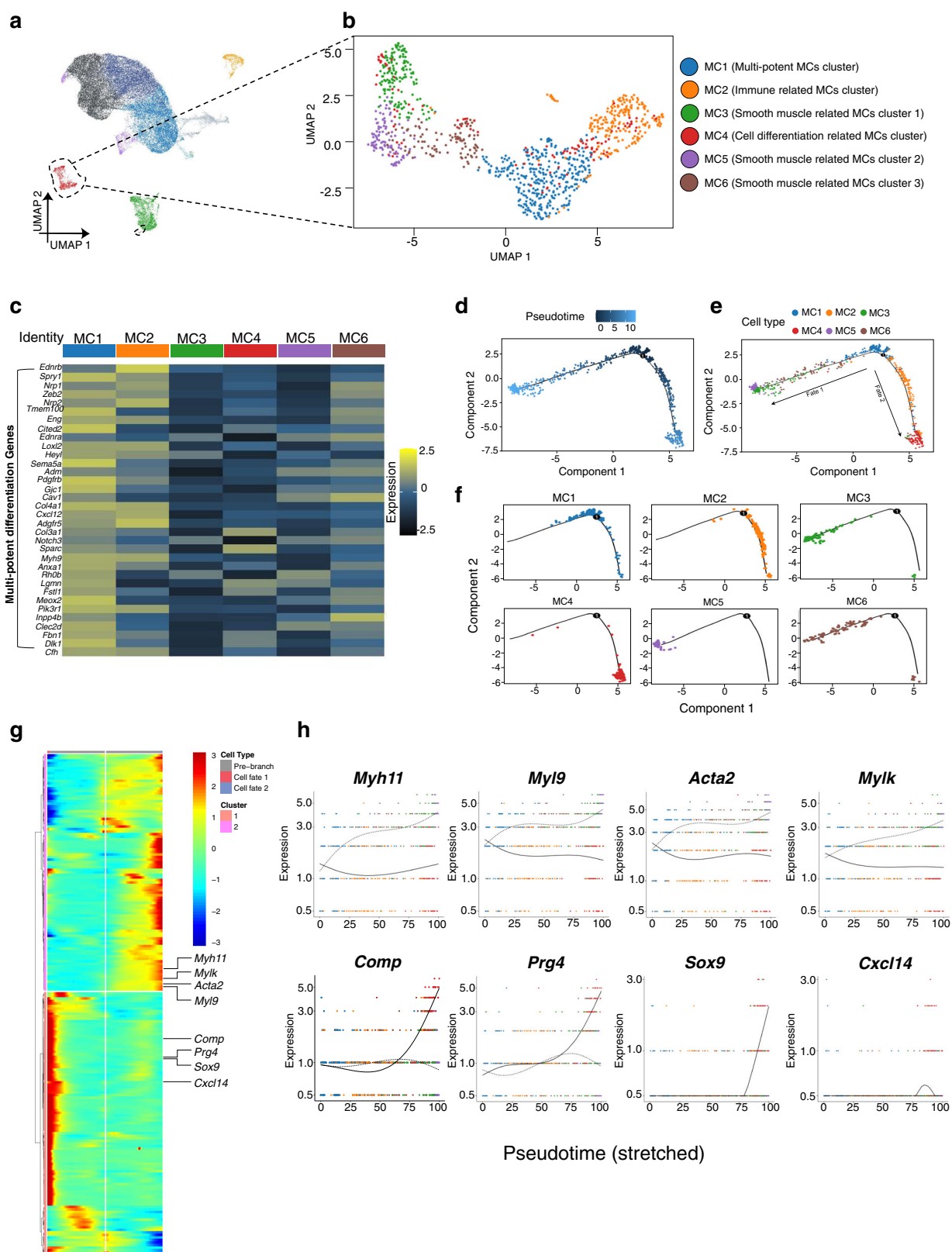

anterior attachment structure is damaged by inflammation and abnormal local pressure, and adhesion and fibrosis appear in the disc, which results in progressive disc degeneration[2,3]. The function of individual cells and tissues is dependent on their structure. Cellular zonation can reflect the division of labor among cells. In TMJ discs, in contrast to the numerous FBs that are distributed throughout the disc

area, ECs, MCs and MPhs, whose roles in the articular disc previously did not draw extensive attention, mainly reside at the very end of the disc bands and are limited to the junction of the anterior band and attachment. The *Sfrp2*+ FBs, which cells were considered as potential functional cells for disc tissue repair, were also found mainly located in this area. This finding highlights the importance of homeostatic

**Fig. 4 | The mural cell subclusters have two pseudotime lineage directions in mouse TMJ disc. a** UMAP plot of TMJ disc cells. Mural cells (MCs) are marked in red. **b** UMAP plot of reclustered MC subpopulations according to the Methods, MC1: multipotent MC cluster, MC2: immune-related MC cluster, MC3: smooth muscle-related MC cluster 1, MC4: cell differentiation-related MC cluster, MC5: smooth muscle-related MC cluster 2, and MC6: smooth muscle-related MC cluster 3. MCs were annotated using a combination of GO and manual annotation. **c** Heatmap of multipotent differentiation genes in 6 MC subclusters. Heatmap depicting the average gene expression of multipotent differentiation genes associated with the GO terms that were enriched in MC1. Selected GO terms: 'mesenchymal cell differentiation', 'vasculogenesis', 'branching involved in blood vessel morphogenesis', 'artery morphogenesis', 'endothelial cell migration', 'smooth muscle cell differentiation negative', 'regulation of osteoclast differentiation', 'regulation of osteoclast differentiation', and 'vascular associated smooth muscle cell differentiation'. **d** Trajectory order of the MC populations by pseudotime value. **e** Monocle analysis of MCs revealed developmental bifurcation. **f** Deconvolution of the Monocle pseudotime plot according to MC subcluster. **g** The expression of genes in a branch-dependent manner. Each row indicates the standardized kinetic curve of a gene. The center of the heatmap shows the kinetic curve value at the root of the trajectory. From the center to the right of the heatmap, the kinetic curve progresses from the root along the trajectory to fate 1. Starting from the left, the curve progresses from the root to fate 2. **h** Pseudotime kinetics of the indicated selected genes from the root of the trajectory to fate 1 and fate 2.

anterior attachment for disc function and improve the understanding of the vascular cell niche within the anterior band attachment area. More importantly, these findings suggest that this area could be a critical site for early clinical diagnosis and for specific target therapy of TMJ disc disarrangement.

Recent and ongoing work has highlighted the important interplay among aging, inflammation, and loss of regenerative potential in multiple tissues. In the TMJ, even disc injuries sustained in youth remain difficult to repair, giving rise to the degeneration of fibrocartilaginous tissue that can lead to accelerated OA pathology. In recent decades, multiple studies have explored putative cartilage progenitor cells (CPCs) in articular cartilage by characterizing their cell surface markers and describing their function[55,56]. In studies of TMJ condylar fibrocartilage, our group and other researchers also found multipotent fibrocartilage stem cells in humans and animals[40,51]. However, whether there are similar progenitor populations residing in the articular disc and how these progenitors are effectively activated for proper function still needs clarification.

The disc atlas presented here described for the first time the existence of resident MCs in the disc bands. MCs are seen as a critical perivascular source for mesenchymal stem cells in multiple organs[23,54,57–59]. In TMJ discs, we found that MCs have typical skeletal progenitor characteristics, with specific transcriptional markers such as *Notch3* and *Thy1*, which both provide identity and mediate the multipotent capacities of this population. Progenitor differentiation requires orchestration of dynamic gene regulatory networks during stepwise fate transitions. In our disc atlas, we found that NOTCH3 and THY1 signaling contribute substantially to the cell–cell signaling cascade in MCs at all stages. The absence of these signaling pathways could lead to loss of the multipotent capacity of progenitor cells. As an example, THY1+ cells in TMJ discs showed markedly stronger stem cell characteristics than THY1- disc cells in vitro and ex vivo. In addition, lineage tracing in vivo showed that *Myh11-CreER*+ MCs proliferated and migrated to coordinate angiogenesis in disc injury, but they gradually lost their NOTCH3 and THY1 expression, as well as progenitor characteristics and ultimately became fibroblasts. Interestingly, the *Myh11-CreER*+ MCs appears to differentiate toward *Sfrp2*+ non-chondrogenic fibroblasts instead of *Chad*+ chondrogenesis related disc cells. This phenomenon, as well as our in vitro results showing that it is hard for MCs to form mature cartilage without chondrogenesis induction, suggests a limited chondrogenesis capacity of MCs for disc fibrocartilage repair. On the other hand, we also observed appreciable *Sfrp2*+/ *Myh11*-RFP- cells being activated on the anterior band. Since *Sfrp2* is critical for regulating progenitor differentiation during tissue regeneration and repairing[60–62], we consider there is probable other resident progenitor source that participate in the regulation of cellular homeostasis together with MCs in the TMJ disc.

NOTCH signaling pathway was found with a dual role during joint cartilage metabolism, as well as being identified as a potential regulator of both catabolic and anabolic molecules in the cartilage ECM during development[63–65]. Transient activation of NOTCH signaling in postnatal chondrocytes results in increased synthesis of cartilage ECM

and joint maintenance, while overexpression of NOTCH signaling activates the pathway in OA cartilage[66]. In TMJ, our previous study showed that NOTCH signaling was excessively activated during the onset and development of TMJOA, while partial blocking NOTCH signaling by preventing NICD release could alleviated the cartilage destruction[67]. In addition, NOTCH activation was found critical for chondrogenic progenitor specifications in condylar fibrocartilage[68]. These findings, in addition to our current study showing that NOTCH is active at all postnatal stages in TMJ disc, suggest potential functions of NOTCH both as the 'identity' and the regulator of MCs characteristics during TMJ cartilage development and injury repair. Our work,

However, there are several limitations of our work: the mechanisms underlying disease susceptibility and the pathogenic processes of cells in the junction are not fully understood. Animal models that activate/eliminate MCs from TMJ discs would provide more direct evidence to elucidate their functional roles during disc development and homeostasis. Moreover, artificially overexpressing/inhibiting the NOTCH and THY1 signaling pathways in MCs during TMJ disc injury would provide a model for developing therapeutic strategies for TMD treatment in the future.

Our work here details an atlas of cell populations, transcriptomes, and inferred interactions, provides insights into cell functionality in postnatal TMJ disc development and homeostasis, and presents for the first time the resident progenitor cells in the articular disc. Our work opens avenues for further mechanistic exploration of the complex process of disc disease and degeneration.

# Methods
## Ethics approval information
This study was performed in strict accordance with the recommendations in the Guide for the Care and Use of Laboratory Animals of Sichuan University. Animal procedures were performed according to protocols approved by the Animal Ethics Committee of Sichuan University (WCHSIRB-D-2021-231). All animal experiments followed the Animal Research: Reporting of In Vivo Experiments (ARRIVE) guidelines.

## Animals
FVB-Tg(*Myh11-cre/ERT2)1Soff/J* mice (JAX#019079, abbreviated *Myh11-CreER*) and B6.Cg-Gt(ROSA)26Sor^tm14(CAG-tdTomato)Hze/J mice (JAX#07908, abbreviated *Tm^fl/fl*) at the age of 7 weeks old were obtained from Jackson Laboratory. These mice were mated with 7-week-old C57/BL6 mice (Dashuo Experimental Animal Laboratories, Chengdu). To generate tdTomato-conditionally activated mice, *Tm^fl/fl* mice were crossed with *Myh11-Cre^ERT* mice. 8-week-old immunodeficient NOD-SCID mice (C001180, Cyagen Biosciences, China) were used as transplant recipients for renal capsule transplantation of prospective MC progenitor populations. 8-week-old GFP+ mice (C001180, Cyagen Biosciences, China) were used for generation of a parabiosis model with non-GFP mice. All animals were maintained in appropriate environment at 24 °C with 40% humidity in a 12 h light/dark cycle with free access to water and irradiated diet. Wood bedding and igloo covers were provided for

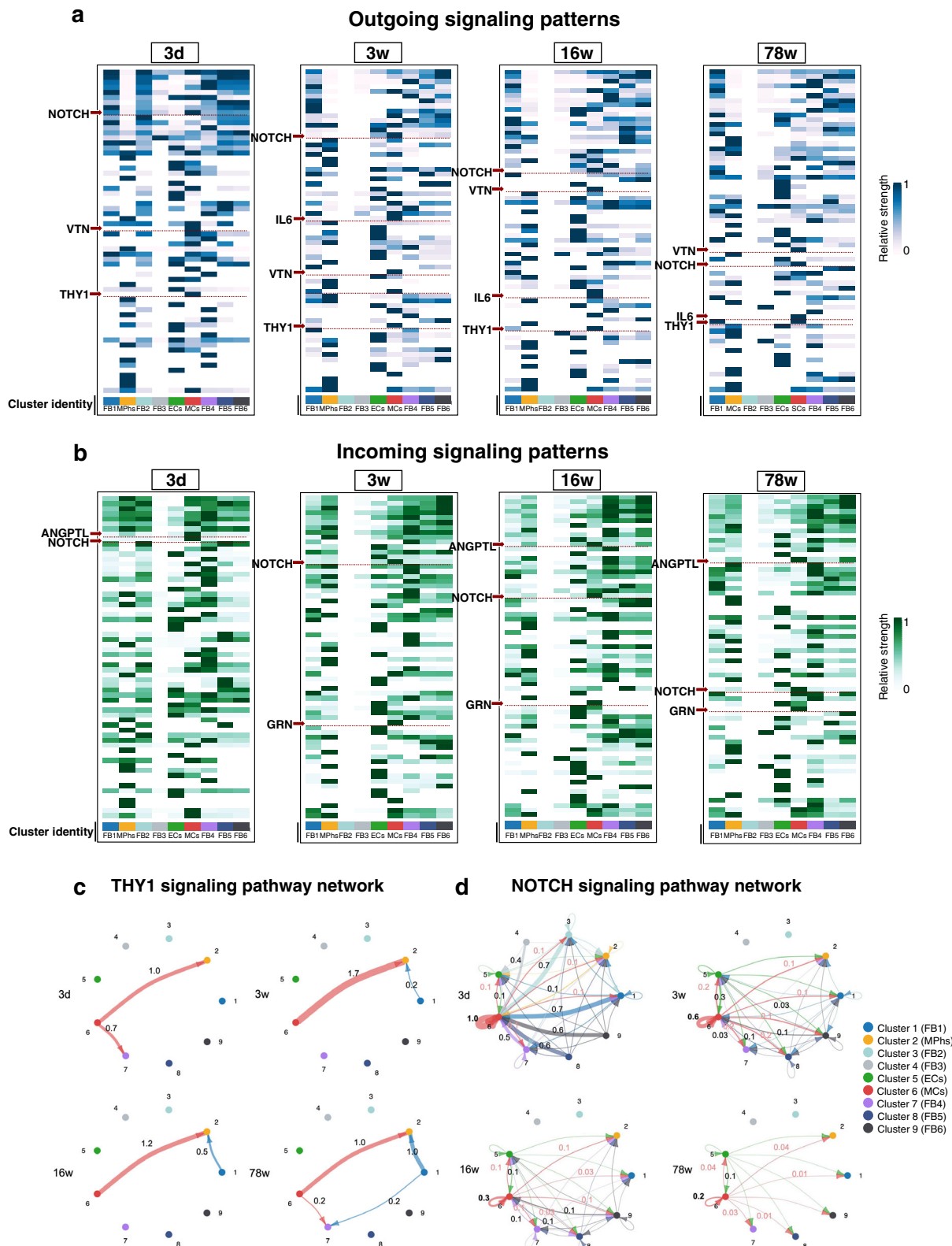

**Fig. 5 | NOTCH/THY1 is activated in mural cells in the postnatal development and aging of TMJ discs. a** Heatmap of active outgoing signaling pathways in different clusters. **b** Heatmap of active incoming signaling pathways in different clusters. The heatmap color bar in (**a**) and (**b**) represents the relative signaling strength of a signaling pathway across cell groups, determined by computing the network centrality scores by Cellchat to find dominant senders (sources) and receivers (targets). **c** Circle plot of the THY1 signaling pathway from CellChat. Receptor-ligand edges were annotated by communication probability. **d** Circle plot of the NOTCH signaling pathway from CellChat. Receptor-ligand edges were annotated by communication probability.

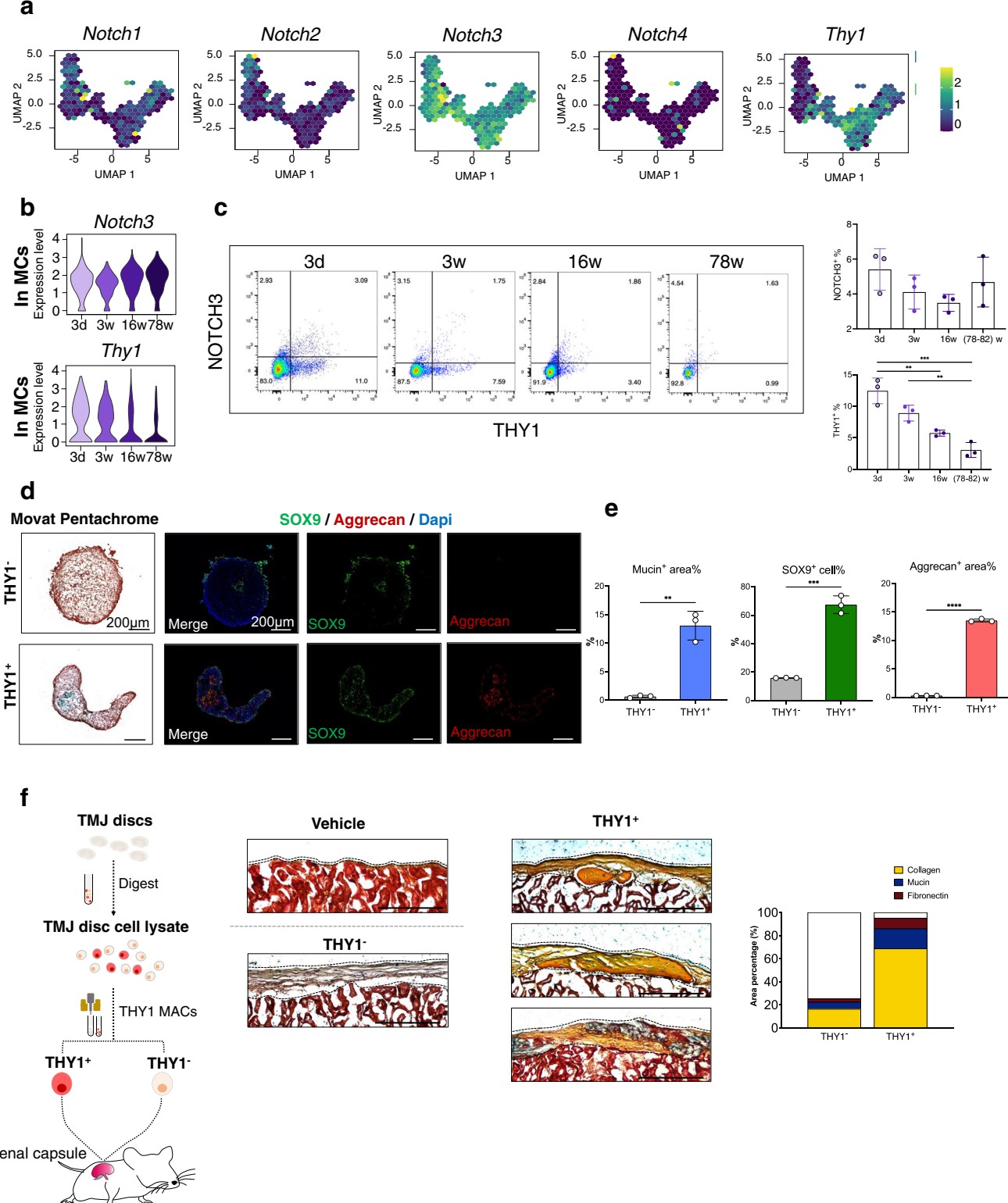

**Fig. 6 | THY1+/NOTCH3+ mural cells in TMJ discs show skeletal progenitor capacities. a** Normalized expression of *Notch* families and *Thy1* visualized in reclustered MCs in low-dimensional space with schex. **b** Violin plot of *Notch3* and *Thy1* at different stages. **c** Flow cytometry plot of THY1+/NOTCH3+ cells (left). % THY1+/NOTCH3+ cells (mean + /-SD) are presented (right). *N* = 3 independent biological samples. The one-way ANOVA with Tukey's multiple comparison test was used for analysis, **(3d vs. 16w) *p* = 0.0014, **(3w vs. 78-82w) *p* = 0.0033, ***p* = 0.0033. **d** Pentachrome staining: Black: nuclei/elastic fibers, bright blue: mucin, and bright red: fibrin. Immunofluorescence staining: red: ACAN; green:

SOX9; blue: DAPI. *N* = 3 biological replicates. Scale bar = 50 μm. **e** %Mucin+ area, %Aggrecan+ area and %SOX9+ cells of THY1+/THY1- chondrogenesis pellets were quantified using ImageJ 1.51. Data are presented as mean + /-SD. *N* = 3 independent animals. The two tailed *t* test was used for data analysis, **p* = 0.001, ***p* = 0.001, ****p* < 0.0001. **f** Movat pentachrome staining of progenitor-derived grafts from explanted discs after renal subcapsular transplantation into NOD-SCID mice; yellow: collagen fiber, blue: mucin, and red: myofiber/fibrin. %Mucin+ area, %collagen+ area and %fibronectin+ area from THY1+/THY1- grafts were semi-quantified using ImageJ 1.51. Scale bar = 200 μm.

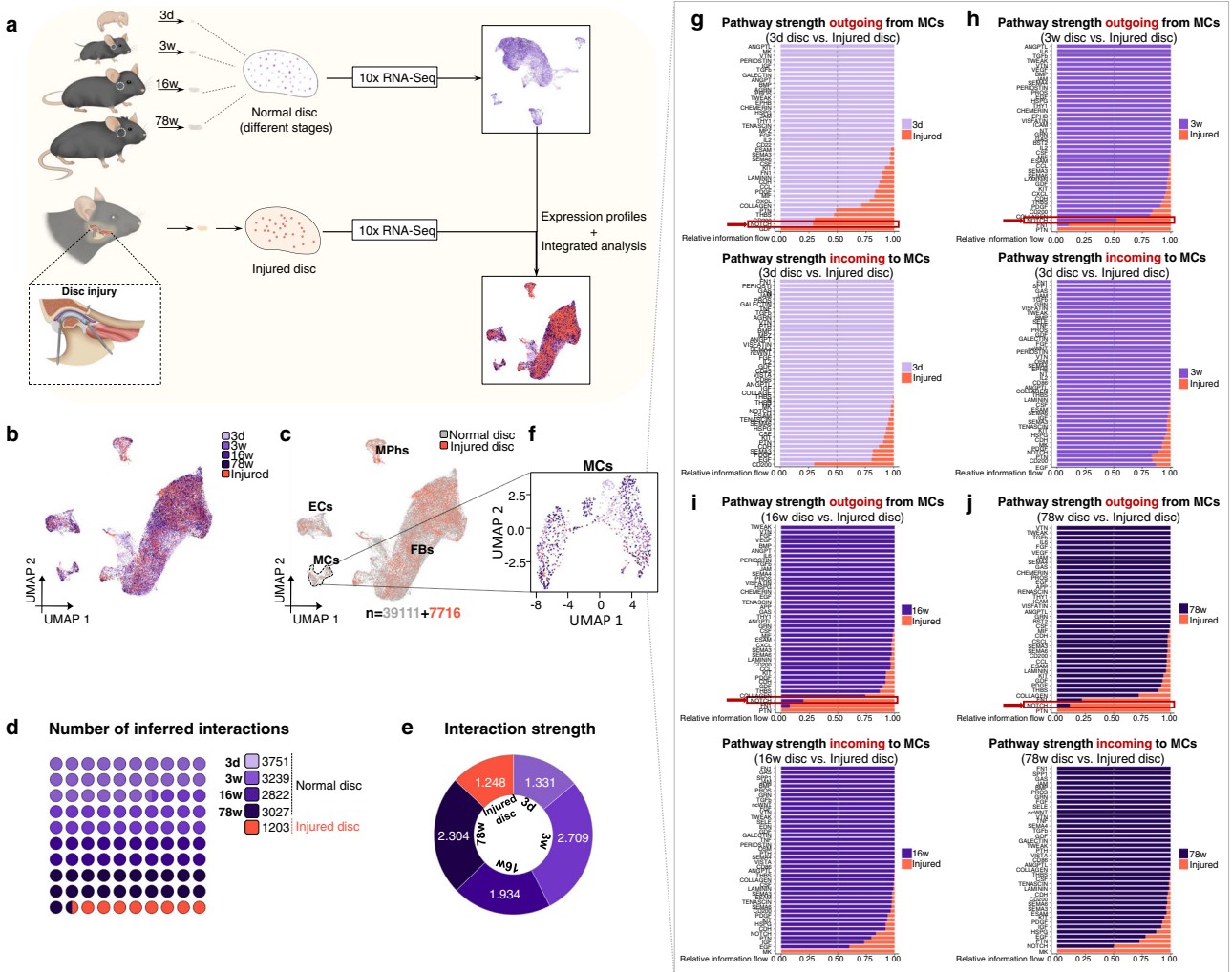

**Fig. 7 | NOTCH pathway is the target signaling for MCs expressional and functional changes after disc injury. a** Schematic of the TMJ disc injury mouse model and integrated single-cell transcriptomic analyses using both normal and injured disc cells. **b** Dimension reduction presentation (*via* UMAP) of integrated single-cell transcriptome data for all disc samples. Each dot represents a single cell and is labeled by different color depths according to stage and injury condition. **c** Dimension reduction presentation (*via* UMAP) of integrated single-cell transcriptome data for normal disc cells (in grey) and injured disc cells (in red). Each dot represents a single cell, the retained cell number is 46827. **d** Proportion of inferred signaling interaction numbers in normal and injured TMJ disc cells. **e** Proportion of signaling interaction strength in normal and injured TMJ disc cells. **f** UMAP plot of

MC subpopulations according to the Methods. Each dot represents a single cell and is labeled by different color depths according to stage and injury condition. **g** Stacked bar graph of outgoing and incoming information flow for each signaling pathway associated with mural cells of 3d and injury TMJ disc datasets. **h** Stacked bar graph of outgoing and incoming information flow for each signaling pathway associated with mural cells of 3w and injury TMJ disc datasets. **i** Stacked bar graph of outgoing and incoming information flow for each signaling pathway associated with mural cells of 16w and injury TMJ disc datasets. **j** Stacked bar graph of outgoing and incoming information flow for each signaling pathway associated with mural cells of 78w and injury TMJ disc datasets.

environmental enrichment. Mice were fed a standard chow diet. All the male/female mice were wild type mice selected from different litters randomly. Both male and female 3-day pups were used for one bulk single cell suspension. 3-week, 16-week and 78-week mice used for all the experiments were males. Since the transgenic locus is located on the Y chromosome, only male mice (Myh11-Cre$^{ER}$; Tm$^{fl/-}$) were used for MCs lineage tracing.

## Single-cell digestion and sequencing

To isolate TMJ disc cells, the discs were freed from the joint space and isolated precisely from the surrounding tissue using fine forceps and scissors under a stereomicroscope. Next, TMJ discs were digested in 4 mg/ml Pronase (10165921001, Roche, Switzerland) for one hour, followed by digesting in 2 mg/ml Collagenase P (11213857001, Roche, Switzerland) in fetal bovine serum (FBS)-free DMEM for 2 h. After filtering the cell suspension using 70 μm and 40 μm cell filters, the cell

suspension was loaded into Chromium microfluidic chips with 3' chemistry and barcoded with a 10x Chromium Controller. RNA from the barcoded cells was subsequently reverse-transcribed, and sequencing libraries were constructed with reagents from a Chromium Single Cell 3' v2 reagent kit according to the manufacturer's instructions. Sequencing was performed with an Illumina system.

## Cluster generation and labeling

Integrated analysis was performed using the Seurat package according to previous works[69,70]. Briefly, count files for each condition were read into RStudio (version 1.4.1106), and datasets corresponding to each sample were labeled accordingly as '3 d', '3 w', '16 w' and '78 w'. Only cells found to express more than 200 transcripts were considered to limit contamination from dead or dying cells. According to the instructions of the Seurat package, each dataset was normalized for sequencing depth with the 'NormalizeData' function, and the 2000

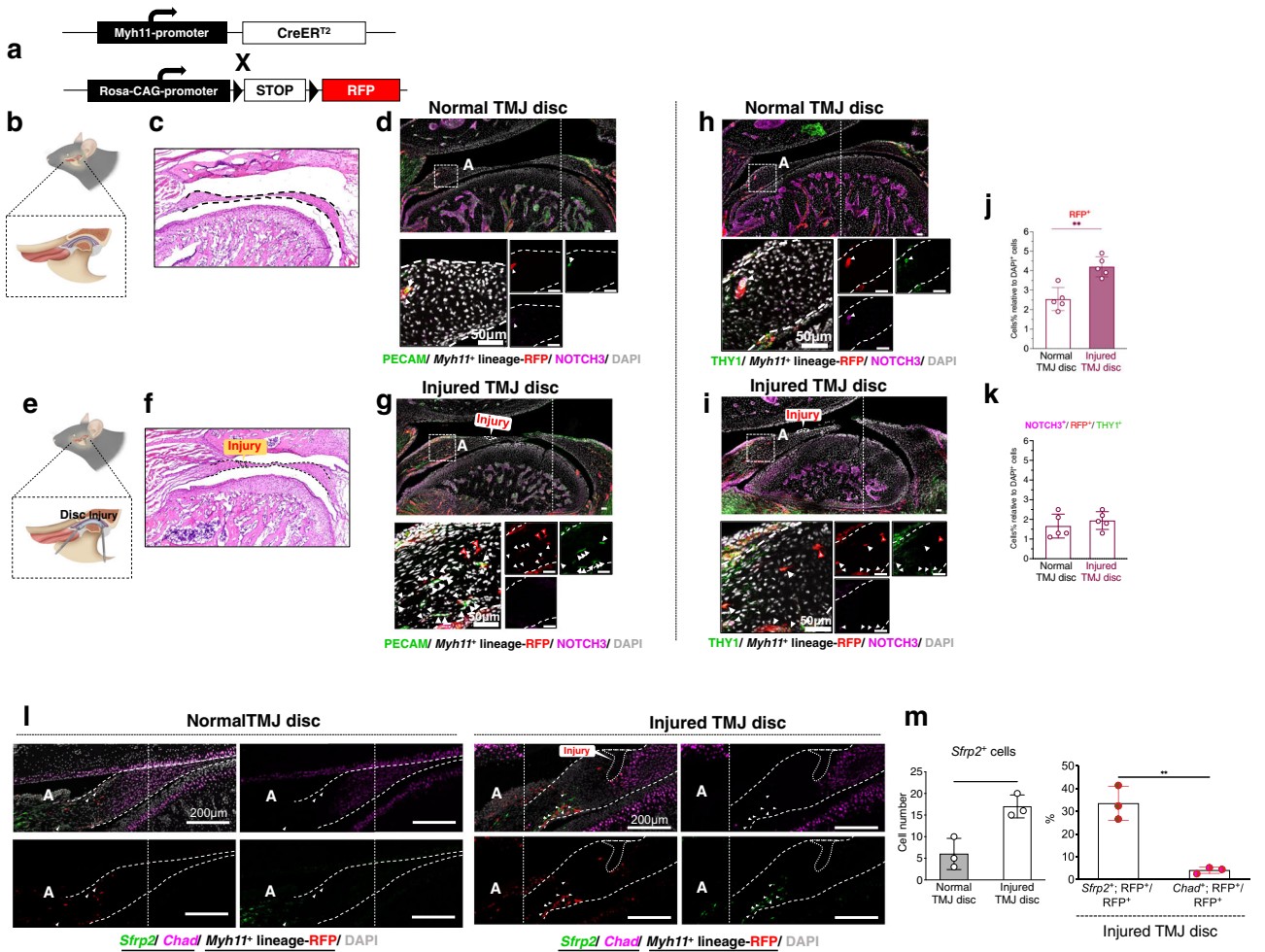

**Fig. 8 | Mural cells transform toward non-chondrogenic fibroblasts in TMJ disc injury. a** Schematic diagram of the mating strategy for generating *Myh11-CreER*; *Tm^{fl/-}* mice. **b** Schematic diagram of the normal anatomical structure of mice TMJ disc. **c** H&E staining of normal discs. Dotted lines: disc boundary. *N* = 5 independent animals. Scale bar:200 µm. **d** Immunofluorescence staining of normal discs. Green: PECAM-1; Magenta: NOTCH3; red: *Myh11*⁺ lineage-RFP; blue: DAPI. Triangular arrows: *Myh11*⁺ lineage; long-tailed arrows: PECAM-1⁺ cells. A: anterior band; dotted lines: disc boundary. *N* = 5 independent animals. Scale bar: 50 µm. **e** Schematic diagram of the anatomical structure of injured discs in mice. **f** H&E staining of injured discs. Dotted lines: disc boundary. *N* = 5 independent animals. Scale bar: 200 µm. **g** Immunofluorescence staining of injured discs. Green: PECAM-1; Magenta: NOTCH3; blue: DAPI. White triangular arrows: *Myh11*⁺ cell lineage; white long-tail arrows: PECAM-1⁺ cells; red triangular arrows: RFP⁺/NOTCH3⁻. A: anterior band, dotted lines: disc boundary. *N* = 5 independent animals. Scale bar: 50 µm. **h** Immunofluorescence staining of normal discs. Green: THY1; Magenta: NOTCH3; red: *Myh11*⁺ lineage-RFP; blue: DAPI. White triangular

arrows: RFP⁺/THY1⁺/NOTCH3⁺ cells. A: anterior band, dotted lines: disc boundary. *N* = 5 independent animals. Scale bar: 50 µm. **i** Immunofluorescence staining of injured discs. Green: THY1; Magenta: NOTCH3; red: *Myh11*⁺ lineage-RFP; blue: DAPI. White triangular arrows: RFP⁺/THY1⁺/NOTCH3⁺ cells; white long-tailed arrows: RFP⁺/THY1⁺/NOTCH3⁻ cells; red triangular arrows: RFP⁺/THY1⁻/NOTCH3⁻ cells. A: anterior band, dotted lines: disc boundary. *N* = 5 independent animals. Scale bar: 50 µm. **j** Semi-quantification of *Myh11*⁺ lineage-RFP⁺ cell percentage. Data are presented as mean +/-SD. *N* = 5 independent animals. The two tailed *t* test was used for data analysis, **p = 0.0015. **k** Semi-quantification of RFP⁺/THY1⁺/NOTCH3⁺ cell percentage. Data are presented as mean +/-SD. *N* = 5 independent animals. The two tailed *t* test was used for data analysis. **l** Combined fluorescence in situ hybridization and immunostaining of inured TMJ discs. Red: *Myh11*⁺ lineage-RFP; Green: *Sfrp2*; magenta: *Chad*; gray: DAPI. A: anterior band, white dotted lines: disc boundary, scale bar:200 µm. **m** Semi-quantification of *Sfrp2*⁺ cell number, and *Sfrp2*⁺/RFP⁺ cell proportion in injured disc. Data are presented as mean +/-SD. *N* = 3 independent animals. The two tailed *t* test was used for data analysis, *p = 0.0131.

most variable features of each dataset were detected using the 'vst' method with the 'FindVariableFeatures' function. Subsequently, the 'FindIntegrationAnchors' function was used to identify anchors across the datasets, and the 'IntegrateData' function was used to integrate them so that an integrated analysis could be run on all cells simultaneously.

The data were then scaled and central features in the dataset were identified using 'ScaleData', and PCA components were used for an initial clustering of the cells (using 'RunPCA'). Thirty dimensions were used to capture the majority of the variability across the datasets. 'FindNeighbors' was then used with the above dimensionality parameters to construct a K-nearest neighbor graph based on Euclidian distances in PCA space. The clusters were then refined by a shared

nearest neighbor (SNN) modularity optimization-based clustering algorithm. This was done by using the 'FindClusters' function. Nine distinct clusters were identified (Clusters 1–9). The clusters were annotated by manual review of the marker genes of each individual cluster. We then performed nonlinear dimensionality reduction using UMAP to visualize and explore the datasets. Gene set over-representation analysis of significantly upregulated cluster-defining genes was performed using 'FindMarkers'. Gene set enrichment analysis was performed using ClusterProfiler[71,72] with GO terms for biological processes.

To identify cell types of interest that were present as smaller populations within the TMJ disc cells, an individual cluster (cluster 6, MCs) was extracted by using the 'Subset' function and subjected to

further subclustering. Six subpopulations of MCs were identified by manual review of the expressed markers and annotated as immune-related MCs, naive MCs, FB-related MCs, and smooth muscle-related MCs in TMJ discs.

## Cell–cell interaction analysis

CellChat[73] is an R-based analysis tool for calculating the interaction between ligands and receptors between different cell populations. We used CellChat for cell–cell communication calculation and analysis in TMJ discs. In brief, overexpressed signaling genes and ligand–receptor interactions (pairs) were found by using 'IdentifyOverExpressedGenes' and 'IdentifyOverExpressedInteractions' within the CellChatDB for five datasets (3 d, 3 w, 16 w, and 78 w, injured). The 'ComputeCommunProb' function was used to compute the communication probability or interaction strengths between any interacting cell groups. To identify the differences between datasets, the 'MergeCellChat' and 'CompareInteractions' functions were used in CellChat. The differential edge list was passed through 'netVisual_heatmap' to generate a heatmap and find receptor-ligand edges between MCs and other cells, and then Circle plots were generated with 'netVisual_circle' in CellChat. Then, we selected several MC-related receptor-ligand edges with a high communication probability for visualization, including the THY1 signaling pathway and NOTCH signaling pathway. In stacked bar graph, different injury specific signaling pathways were identified, by using "rankNet" function to compare outgoing and incoming information flow for each signaling pathway associated with mural cells of normal and injury datasets.

## Pseudotime analysis

To determine the differentiation trajectory of MCs, we used the Monocle2[74–76] package to calculate the differentiation process of these cells. Monocle uses the CellDataSet object to store single-cell gene expression data as well as analysis results. Thus, we created a CellDataSet object for single cells of the integrated dataset after subclustering the MCs cluster (MCs). Genes that provided important information for shaping the trajectory were detected through the 'FindVariableGenes' function in the Seurat package. Then, dimensionality reduction and trajectory construction were performed with the obtained genes. After determining the pseudotime value arrangement and differentiation trajectory, we detected genes that followed different kinetic trends along the different branches from the starting state using the Beam function in Monocle. The genes (selected based on $q$ value<1e-4) of MCs were used to draw a heatmap. Hierarchical clustering was applied to cluster the genes into 2 subgroups according to expression patterns. Additionally, we used the 'plot_cell_trajectory' function in the Monocle2 package to show the changes in cells in the MC subcluster during the differentiation trajectory. Besides, we chose 3d scRNA-seq data and used CytoTRACE (v.0.3.3) to predict differentiation states of different FB subclusters at this stage[46].

## TMJ disc cell isolation and culture

TMJ discs were dissected from 3-day-old C57BL/6 mice and immediately placed into phosphate-buffered saline (PBS) solution at room temperature. TMJ discs were digested in 4 mg/ml Pronase (10165921001, Roche, Switzerland) for one hour, followed by digesting in 2 mg/ml Collagenase P (11213857001, Roche, Switzerland) in fetal bovine serum (FBS)-free DMEM for 0.5–1 h. Two-step enzyme digestion was performed at 37 °C with horizontal shaking at 100 rpm. An equivalent volume of medium was added to terminate the reaction. Then, the cell suspension was passed through a 40 μm cell strainer. Cells were pelleted by centrifugation at $300 \times g$ for 5 min. The supernatant was removed, and the cell pellet was resuspended in magnetic-activated cell sorting (MACS) buffer (PBS supplemented with 0.5% bovine serum albumin and 2 mM EDTA). MACS with CD90.2/THY1 microbeads (130-121-278, Miltenyi Biotec, Germany) was used to

isolate MCs. THY1+ and THY1- MCs were both retrieved after MACS. Cells were cultured with high-glucose Dulbecco's modified Eagle's medium (DMEM) (C11995500BT, Gibco, USA) with 20% fetal bovine serum (FBS) (10091148, Gibco, USA) and 1% penicillin/streptomycin (15140122, Gibco, USA) in a humidified incubator at 37 °C and 5% $CO_2$. The medium was completely changed every 3–4 days. When the cultures reached 75% confluence, the cells were subcultured using 0.25% trypsin-EDTA (25200072, Gibco, USA).

## Flow cytometry (FCM)

TMJ discs were isolated from C57BL/6 mice at different stages (3 d, 3 w, 16 w, and 78-82 w) and then cut into small pieces. Next, these small disc pieces were digested with pronase for 1 h and collagenase P for 0.5–1 h as previously described. The dissociated cells were centrifuged, filtered through nylon mesh, and resuspended in cell staining buffer. FCM was performed using a flow cytometer (AttuneTM NxT Flow Cytometer Thermo Fisher Scientific, USA).

TMJ discs cells were immunolabeled with fluorescent conjugated antibodies on ice for 35 min. For antigens requiring 2-step antibody staining, cells were stained on ice for 35 min with unconjugated primary antibodies followed by fluorophore-conjugated secondary antibodies on ice for another 35 min. Then the cells were washed with FACS buffer, centrifuged for 5 min at $300 \times g$, resuspended and placed on ice.

After blocking with 10% goat serum in 5% BSA, TMJ discs cells were immunolabeled with fluorescent conjugated antibodies on ice for 35 min. For antigens requiring 2-step antibody staining, cells were stained on ice for 35 min with unconjugated primary antibodies followed by fluorophore-conjugated secondary antibodies on ice for another 35 min. Then the cells were washed with FCM buffer, centrifuged for 5 min at $300 \times g$, resuspended and placed on ice.

The fluorescent conjugated antibodies included Alexa flour 647 anti-NOTCH3 (Alexa flour 647, 130512, Biolegend, USA). The unconjugated primary antibodies included rat anti-PECAM-1 (550274, Becton, Dickinson and Company, USA), mousecanti-C1QA (NBP1-51139, NOVUSBIO, USA) and rat anti-THY1 (14-0902-82, Invitrogen, USA). The fluorophore-conjugated secondary antibodies included Alexa fluor 488 goat anti-rat (A23240, Abbkine, USA) and Alexa fluor 488 goat anti-mouse (A32723, Invitrogen, USA). The sample gating strategies for all FCM experiments were described in supplementary Fig. S12.

## 5-Ethynyl-2'-deoxyuridine (EdU) labeling assay

C0071 BeyoClickTM EdU-488 was used to measure the proliferation capacity of THY1+ and THY1- cells. Cells at P2 were seeded in 24-well plates at $2 \times 10^4$ cells/well. When the cells reached 50% confluence, 10 μM EdU was added to the medium, and the cells were cultured for 4 h. The cells were then fixed with 4% paraformaldehyde (PFA) and incubated with Click Additive Solution. The nuclei were counterstained with 4',6-diamidino-2-phenylindole (DAPI) (C0065, Solarbio, China). Positive cells were observed and counted under a fluorescence microscope (DMi8, Lecia, Germany). The percentage of EdU-positive cells was calculated by the number of green fluorescent cells/number of DAPI-stained cells.

## Colony formation assay

THY1+ and THY1- cells at P2 in the logarithmic phase were harvested and cultured (1000 in a 6-well plate). After culture for 12 days, the cells were fixed with 4% paraformaldehyde for 15 min, washed in PBS and stained with 0.1% crystal violet dye (G1072, Solarbio, China) for 2 min. The cells were next washed with $ddH_2O$ until the rinsing water was clear, and the dishes were air dried and photographed.

## Multilineage differentiation

Multilineage differentiation was investigated in vitro using chemically defined medium for inducing chondrogenesis, osteogenesis, and

adipogenesis. For chondrogenesis, THY1$^+$ and THY1$^-$ cells ($2.5 \times 10^5$) at P2 were pelleted in 96-well plates by centrifugation and cultured for 3 weeks in high-glucose Dulbecco's modified Eagle's medium (DMEM) (C11995500BT, Gibco, USA) supplemented with $10^{-7}$ mol dexamethasone (D4902, Sigma, USA), 1 mM sodium pyruvate (P4562, Sigma, USA), 50 μg/ml L-ascorbic-2-phosphate (A4403, Sigma, USA), 40 μg/ml l-proline (P5607, Sigma, USA), 1% insulin, transferrin, selenium (ITS, 12521, Sigma, USA), and 10 ng/ml TGF-β3 (100-36E-10, PeproTech, USA). Cells were centrifuged for 5 min at 300 $g$ to form a pellet. Three weeks later, the pellets were fixed in 4% paraformaldehyde and prepared for frozen embedded sections and were stained with Movat pentachrome (ab245884, Abcam, UK). To induce osteogenesis, THY1$^+$ and THY1$^-$ cells ($2 \times 10^4$) at P2 were cultured in 24-well plates for 1 week in medium containing αMEM supplemented with 10% FBS, $10^{-8}$ mol dexamethasone, 50 μg/m L-ascorbic acid, and 10 mM β-glycerophosphate (G8100, Solarbio, China). After 2 weeks, the cells were fixed in 4% PFA and stained with 1% Alizarin red (G1452, Solarbio, China). To induce adipogenesis, THY1$^+$ and THY1$^-$ cells ($2 \times 10^4$) at P2 were cultured in 24-well plates using adipogenesis-induced medium A: αMEM with 20% FBS, 500 μM IBMX (I7018, Sigma, USA), 1 μM dexamethasone, 1 μM rosiglitazone (R8470, Solarbio, China) and 10 μg/ml insulin (I8040, Solarbio, China). 48 h later, the cells were washed with PBS and cultured using induction medium B: αMEM with 20% FBS, 10 μg/ml insulin, and 1 μM rosiglitazone. Medium B was completely changed every 2 days. After 6 days of induction, oil red O staining was performed.

## RNA Isolation and qRT-PCR
Total RNA of THY1$^+$ cells and THY1$^-$ cells was purified from 3d TMJ disc with RNeasy Micro Kit (74004, Qiagen, Germany) and treated with DNase I (AM2222, Ambion, USA) to remove genomic DNA. RNA quantity and purity were determined using spectrophotometer (DeNovix, Inc.). RNA samples ($260/280 \geq 1.8$) were used to obtain cDNA (4374967, Invitrogen, USA). qRT-PCR was performed using SYBR Green PCR Master Mix (A25780, Applied Biosystems, USA) and mouse primers of *Ppar-γ*, *Ocn* and *Acan* (Integrated DNA Technologies). Gene expression levels were normalized to the housekeeping gene Glyceraldehyde 3-phosphate dehydrogenase (*Gapdh*).

## Preparation of TMJ disc tissue
The TMJ samples were collected using a scalpel and forceps and immediately fixed with 4% PFA at 4 °C overnight. Samples were decalcified in 446 mM ethylenediaminetetraacetic acid (EDTA) in PBS (pH 7.4) at 4 °C for 1–4 weeks, with a change of EDTA every day. Sucrose (10%, 20%, and 30%) was used for gradient dehydration. Tissues were embedded in optimal cutting temperature (OCT) compound (4583, Sakura Finetek USA, Torrance, CA) or paraffin. Slices with a thickness of 5–7 or 30 μm were made along the sagittal plane.

## Histology
Dissected specimens were fixed in 4% PFA at 4 °C overnight. Samples were decalcified in 446 mM ethylenediaminetetraacetic acid (EDTA) in PBS (pH 7.4) at 4 ˚C for 1–4 weeks with a change of EDTA once every day. 10%, 20%, 30% sucrose were used for gradient dehydration. Tissues were embedded in optimal cutting temperature (OCT) compound (4583, Sakura Finetek USA, Torrance, CA) or paraffin. 5–7- or 30 μm slices were made with in a sagittal plane. Representative sections were stained with Modified Russell-Movat Pentachrome Staining Kit (ab245884, Abcam, UK) or Modified Hematoxylin-Eosin (HE) Stain Kit (G1121, Solarbio, China).

## Immunofluorescence staining
Immunofluorescence was performed on cryo-sections. Tissue sections were thawed at room temperature and washed in PBS three times. Permeabilization was performed with 0.5% Triton in PBS when necessary. Then, the samples were blocked with 5% goat serum in 3% BSA in PBS for 1 h at room temperature. Primary antibodies, which included antibodies against CD90.2 (14-0902-82, Invitrogen, USA) diluted to 1:100 in blocking solution, PECAM-1 (550274, BD Pharmingen, USA) diluted to 1:100 in blocking solution, NOTCH3 (PA519515, Invitrogen, USA) diluted to 1:400 in blocking solution, and C1QA (NBP1-51139, Novus, USA) diluted to 1:200 in blocking solution, were incubated at 4 °C overnight. Specimens were washed with PBS for 3 times on the second day. Secondary antibodies, which included goat anti-rat Alexa fluor 488 (a23240, Abbkine, USA), goat anti-rabbit Alexa fluor 568 (A-11036, Invitrogen, USA), goat anti-rabbit Alexa fluor 647 (HA1123, Huabio, China), and goat anti-mouse Alexa fluor 488 (A32723, Invitrogen, USA) antibodies (diluted to 1:500 in blocking solution), were applied for 1 hour. We were careful to wash sections in PBST (1% Tween 20 in PBS) twice and PBS to remove unbound secondary antibodies. The samples were incubated with mounting medium (with DAPI) (S2110, Solarbio, China) for 10 min and mounted with coverslips (80340-3610, CITOTEST, China). Tissue sections were visualized with a laser scanning confocal microscope (LSCM) (Olympus FV3000, Japan).

## Immunohistochemistry
Immunohistochemistry was performed on paraffin sections. The TMJ samples were collected with forceps and scissors and fixed in 4% PFA at 4 °C overnight. All specimens were put into a Leica ASP300S for dehydration after being washed in PBS.

The protocol was performed according to the manufacturer's instructions for the immunohistochemistry staining kit (SP-9000, ZSGB-BIO, China). Samples were deparaffinized with xylene and rehydrated with decreasing concentrations of ethanol. Heat-based antigen retrieval was performed at 95 °C. After the sections cooled, solution 1 was added to block endogenous peroxidase activity. Then, the sections were incubated with blocking solution before being individually incubated at 4 °C overnight with a primary antibody against collagen I (Col I, 1:100, ab34710, Abcam, UK). Subsequently, the samples were incubated with solution 3 and solution 4 for 20 min each, and then 3,3-diaminobenzidine tetrahydrochloride (DAB) substrate was used for color development. Sections were dehydrated in a continuous ethanol and sealed with neutral balsam (BL704A, Biosharp, China). The results were visualized with an optical microscope (Leica DM2000 & DM2000 LED, Germany).

## Fluorescence in situ hybridization (FISH)
The TMJ samples were collected using a scalpel and forceps and immediately fixed with 4% PFA at 4 °C overnight. All specimens were decalcified in Preserve decalcifying solution (0500-0052, Pursuit Bio, China) at 4 °C for 20 h. 10%, 20%, 30% sucrose were used for gradient dehydration. Tissues were embedded in optimal cutting temperature (OCT) compound (4583, Sakura Finetek USA, Torrance, CA). 15 μm slices were made with in a sagittal plane. Then FISH was performed with the user manual of RNAscope Multiplex Fluorescent Reagent Kit (323100-USM, Advanced Cell Diagnostics, USA). Anti-RFP antibody (ET1704-21, Huabio, China) was used to labeled RFP whose fluorescence signal may be diminished during sample processing. Fluorescence was detected with a confocal microscope (Olympus FV3000, Japan).

## Subcapsular renal transplantation of cells
A total of 10000-20000 THY1$^+$ and THY1$^-$ disc cells were sorted using MACS with CD90.2/THY1 microbeads (130-121-278, Miltenyi Biotec, Germany, described in **TMJ disc cell isolation and culture**). Then, the cell pellet was resuspended in 4 μl Matrigel (365230, Corning, USA) on ice. A Wiretrol II syringe (5–000-2010, Drummond Scientific, USA) was assembled to aspirate cell-laden Matrigel. Then, the syringes were set aside at room temperature to let the Matrigel solidify. Eight-week-old adult immunodeficient NOD-SCID mice (C001180, Cyagen

Biosciences, China) were anesthetized by intraperitoneal injection with tribromoethanol (T903147, Macklin, China). Routine preparation and surgical draping of each animal around the surgical site included shaving, preparing the midline dorsal skin with 70% ethanol and povidone-iodine solution, draping and surgical site exposure. An incision was made after palpating the kidney under the skin. Then, the skin was separated from the abdominal wall using blunt dissections. An incision less than 1 cm was made into the abdominal wall, and the kidney was gently manipulated until it protruded through the cut. The kidney poles were identified, and a shallow 1–2 mm incision in the capsule was made with a 31-gauge needle (DS195-21-23, Narang Medical, USA). Pipette tips were used to gently separate the capsule from the renal parenchyma, and then cell-laden Matrigel was injected under the renal capsule using the loaded syringe. After cell implantation, the renal capsule was closed with simple interrupted 12-0 sutures. The abdominal wall and skin were closed with simple interrupted 6-0 sutures (F601, Jinhuan Medical, China). Four weeks later, the mice were euthanized for further Movat pentachrome staining (ab245884, Abcam, UK).

### Temporomandibular joint (TMJ) disc injury model

Five-week-old *Myh11-Cre^ER+; Tm^fl/-* mice were used to determine the effect of MCs on injured TMJ discs. Mice were anesthetized using the intraperitoneal injection of tribromoethanol. The right cheek skin was prepared and sterilized with 70% ethanol and povidone-iodine solution. A vertical incision of less than 1 cm was made around the end of the zygomatic arch. Blunt tools were used to resect the muscles to expose the TMJ articular capsule. At a distance of 0.25 mm from the edge of the anterior bands of the TMJ disc, a 31 G needle was used to puncture the anterior bands three times at each side, with 0.3 mm spacing between adjacent puncture holes. Muscle and skin were sutured with 6-0 Prolene sutures (DePuy Synthes, USA). Two weeks later, the animals were sacrificed for further analyses.

### Parabiosis

The parabiosis model was generated according to previous reports[77,78]. Briefly, each GFP+ female mouse was placed in the same cage with another C57BL/6 non-GFP female mouse for at least two weeks. Mice were anesthetized using an i.p. injection of tribromoethanol. The TMJ disc injury model was established in the right TMJ of non-GFP mice prior to parabiosis according to the **TMJ disc injury model**. Animals were placed in the supine position, with the GFP+ mouse on the right and the non-GFP mouse on the left. After shaving and preparing the midline dorsal skin with 70% ethanol and povidone-iodine solution, a longitudinal skin incision was made on the shaved sides of each animal from 0.5 cm above the elbow to 0.5 cm below the knee joint. Following the incision, the skin was gently detached from the subcutaneous fascia by holding the skin up with a pair of curved forceps and separating the fascia with a second pair to create 0.5 cm of free skin, and this separation was performed along the entire incision. The right olecranon of GFP+ mice and the left olecranon of non-GFP mice were sutured together with 6-0 sutures. Similarly, the knee joints of these two mice were joined together. To better establish circulation chimerism, a simple interrupted suture was performed around the abdominal wall and thoracic wall. Then, the skin of the two animals was connected with 6-0 sutures. Four weeks later, the parabiosis pairs were sacrificed for further studies. Livers were dissected to confirm chimerism, and TMJ tissues were dissected for further analyses.

### Tamoxifen administration

In the TMJ disc injury model, *Myh11-Cre^ER+; Tm^fl/-* mice were injected intraperitoneally with tamoxifen at five weeks old (75 mg/kg body weight/day, for 2 consecutive days). Two days after the first tamoxifen injection, TMJ injury or sham injury was generated in mice. Ten days

after the first tamoxifen injection, the mice were sacrificed for histological analysis.

### Statistics and reproducibility

All data used in the present study were generated and designed by the original studies in which they appear. No statistical method was used to predetermine sample size. No data were excluded from the analyses. Every quantification was measured by 2 investigators individually and blindly when positive cell numbers and positive staining area were counted. The final statistical results were based on the measurements from these 2 investigators. The staining results were analyzed by ImageJ (version 1.51 Leeds Precision Instruments, USA). The FCM results analysis were proceeded by using FlowJo (version 10.6.2 Tree Star, USA). All statistics were calculated using Prism 9 (GraphPad Software). The statistical significance of differences between 2 groups was determined using unpaired Student's $t$ test, and the statistical significance of differences among 3 or more groups was determined using one-way ANOVA with Tukey's multiple comparisons. The statistical difference of the cell growth curve (Fig. 6e) was determined using two-way ANOVA with Tukey's multiple comparisons. Differences were considered to be statistically significant when the $p$ value was <0.05.

### Reporting summary

Further information on research design is available in the Nature Portfolio Reporting Summary linked to this article.

## Data availability

The datasets generated during and/or analyzed during the current study are presented in the paper. The cell counting and positive signal area counting data generated in this study are provided in the Source Data files. The scRNA-Seq data generated in this study have been deposited in the Gene Expression Omnibus (GEO) database under accession code GSE218785. Source data are provided with this paper.

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

## Acknowledgements

This study was supported by National Natural Science Foundation of China (NSFC) No. 82071139, 81771097 (S.Z.), 82270999 (R.B.); Key R&D Program of Sichuan Provincial Department of Science and Technology No. 23ZDYF2130 (to S.Z.); 'From Zero to One' Innovative Research Program of Sichuan University No. 2022SCUH0022 (R.B.). We would like to thank Dr. Zhiqi Zhang, from Sun Yat-sen University First Affiliated Hospital, for his help on generating protocols for TMJ disc digestions for scRNA-Seq. We would thank Prof. Shan Chen and her team, from Sichuan Normal University, for their help on drawing the schematic diagrams.

## Author contributions

R.B. and S.Z. designed the study. R.B., Q.Y., H.L., H.F., Q.L. and P.Li. performed the animal studies. H.L. and Y.W implemented the in vitro experiments using primary cultured disc cells. X.Y. performed all computational analyses. Y.F. and B.Y. interpreted biological signals and guided Q.Y., H.L., H.F., Q.L. and P.Li on highlighting biological insights. Y.F. and P.Ly. provided guidance on transgenic mouse breeding and lineage tracing strategies. All the authors wrote and reviewed the manuscript.

## Competing interests

The authors declare no competing interests.
