## [Peer Review File · Nature Communications]

A single-cell transcriptional atlas reveals resident progenitor cell niche functions in TMJ disc development and injuryREVIEWER COMMENTS

Reviewer #1 (Remarks to the Author):

This study from the Zhu lab aims to delineate a comprehensive roadmap of TMJ disc cells at various post-natal ages in mice. The study team uses single-cell RNAseq to identify 4 cell types, including fibroblasts, endothelial cells, macrophages and mural cells, with 9 distinct clusters. Most notably, NOTCH3+/THY1+ MCs were identified to be progenitor cells that self-renew and have multi-potent ability both in vitro and in vivo. The authors use a TMJ disc injury model and lineage tracing to argue that MCs differentiate into other disc cell types during repair.

Overall this is an important study that addresses a critical medical problem. TMJ biology is severely understudied and treatment for TMDs are limited. The sequencing data provides key information that would advance the understanding of heterogenous TMJ disc cell types and thus represents a significant advance to the field. The computational analyses are comprehensive. However there are several weaknesses that need to be addressed.

The team uncovered 4 cell types, whereas 2 cell types were previously historically known as fibroblasts and chondrocyte-like cells. However it is unclear whether chondrocytes were discovered in their analyses as chondrocytes are not mentioned. This is an important point to make in defining cell types and distinguishing the disc from the condyle and also for reconciling previous studies to this new scRNAseq data.

The majority 88% of the disc population are fibroblasts. This population is highly studied in regenerative medicine/tissue engineering. While a section of the study describes the 6 FB clusters, the authors missed the opportunity to clearly define and name those FB populations, their markers, and describe their speculated functions. Furthermore, since multiple time points are collected authors should expand on how they propose each FB population contributes to disc development, maintenance and possible age-dependent disease. A proposed model and/or table of the different FB populations at varying timepoints and their putative functions would be highly valuable to the field.

The description of mural cells as both a niche and as progenitor cells is a bit confusing. Do the authors mean perivascular niche? If the authors argue MCs are niche cells, which cells do they provide a niche for? Please clarify.

The title reads a bit awkward. It's unclear if the authors are referring to the perivascular location as the niche or the NOTCH3+/THY1+ MCs themselves. NOTCH3+/THY1+ MCs as both a niche cell and progenitor cell is confusing. MC/progenitor cell function during disc development was not directly validated in this study, which would require loss of function studies. Unless authors can validate developmental function during development, its recommended to remove from the title.

In Figure 3c and 3f, immunostainings for PCAM1 and C1QA are shown separately and not co-localized as described in results section line 552. TMJ disc tissue is quite thin and cell constituents can vary dependent on different sections. Its difficult to conclude ECs and Macrophages work together in the anterior band without showing the two populations in the same section. It seems that PCAM1 is localized solely in the disc lining the superior joint cavity, whereas C1QA is found in both super/inferior joint cavity. How does this specific location correlate to their proposed function and relationship to each other?

Figure 4 is very interesting. However, what type of "functional disc cells" do MCs become? Please define "functional cell". This goes back to the question of what is a disc FB?

In Figure 5i-5k. The histology and pentachrome/chemical staining for bone and cartilage is not very convincing. In 5i Thy+ pellets seems highly variable in expressing Acan and pellet staining is

unconvincing. In Fig 5K THY+ cells, the cartilage area (presumably blue color) appears to be missing and the lowest panel that shows "bone" looks more like fibrous tissue. It seems likely that THY+ cells have an inclination toward bone but not cartilage. More markers and/or better sections need to be shown to support the conclusions that THY+ are multipotent. To this end, it is unclear how quantification in Fig. 5K of the bone/cartilage/stroma was performed with the unconvincing histology. Please clarify the quantification.

The Parabiosis model is not sufficiently described in the written results section.

The authors should consider using more recent nomenclature and change the term "mesenchymal progenitors" to "skeletal progenitors" to align with current skeletal stem/progenitor cell biology.

In Fig 8: The Myh11-CreER mouse model needs to be validated in the TMJ disc by In situ or IHC for Myh11. Please add pulse/chase timeline to figure since results and interpretation are highly dependent on recombination timing. The disc injury model is new and seems very technical for a small animal model. It's not clear how reproducibility was controlled. Please clarify surgical standardization methods. Moreover, the injury in Figure 8g-8i seems very small and it's unclear how the location of the injury was determined via histology vs tissue artifact. The injury is in the middle and distant from the anterior region. Why was this location chosen over anterior region where most of the MCs/progenitor cells were located? Were the authors testing secreted factors to encourage migration/differentiation to injury site? Please choose another pseudo-color for NOTCH given yellow also denotes overlap of green/red signal.

Fig 5 shows MCs have a propensity to make bone but how does that property relate to their function? Is bone found in the disc in disease or injury? During injury do Myh11-expressing Notch+ Thy1+ cells transition to bone cells? It seems there's a disconnect between the experiments that need some fine-tuning.

The discussion was ok. It seems redundant of the results. More effort should be made to summarize the 4 populations and their respective subpopulations and to provide speculation on their respective functions as it relates to disc tissue development, homeostasis and disease.

While this study focuses on one progenitor cell population among MCs, it seems this is a very small population with not that many progeny per lineage tracing experiment. Thus there are likely other progenitor cell populations that give rise to other mature cell types that need to be discussed here. A discussion of TMJ disc progenitor cells in the context of skeletal stem/progenitor cells would be warranted here and broaden the scope of this work.

Please discuss how this work would improve patient care/diagnoses as suggested in intro. It may also be helpful to discuss how these newly defined cell populations may shift during aging and disease.

Reviewer #2 (Remarks to the Author):

In this manuscript, Bi et al. performed Single-cell RNA-Seq of the TMJ disc and demonstrated that TMJ disc tissue contains 9 distinct clusters and 4 types of cells: fibroblasts, endothelial cells, macrophages and mural cells. The authors showed spatial-temporal heterogeneity of fibroblast cells throughout TMJ disc tissue during postnatal development and aging and determined the role of macrophages and endothelial cells in TMJ development. The authors also demonstrated that the Notch/Thy1 pathway are active in mural cells during TMJ postnatal development and aging and characterized mesenchymal progenitor capacities of Thy1+/Notch3+ mural cells. Overall, this is an interesting study revealing the functional role of 4 types of cells in TMJ disc tissue. However, some questions, listed below, need to be addressed.

1)For Figure 6c, the thresholds of flow cytometry should be kept in consistent. Please repeat this experiment.

2)The authors reported that diameters of the TMJ discs were increased at different postnatal stages. Does the cell size of TMJ disc also increase at different postnatal stages?

3)Please elaborate the relationship between distribution of different fibroblast clusters and ECM features, especially the different types of collagen. It would be interesting to detect different fibroblast clusters by IHC or IF in TMJ disc tissues.

4)Figure 3c and 3d showed that macrophages were gradually decreased comparing the newborn mice to the aged mice. Have the authors analyzed macrophage polarization in this process? It is not clear if M1 and M2 markers are both decreased from the newborn stage to the aged stage.

5)It has been shown that Notch signaling participates in the onset and development of TMJ-OA and inhibition of Notch signaling pathway temporally postpones the cartilage degradation of TMJ arthritis in mice (DOI: 10.1016/j.jcms.2018.04.026). In this study, the authors found that Notch signaling was activated at all postnatal stages in mural cells. Are these mice suffering from spontaneous TMJ-OA? Please discuss more about Notch signaling in TMJ-OA development.

Reviewer #3 (Remarks to the Author):

The paper "Single-cell transcriptional atlas revealed that the resident progenitor cell niche functions in TMJ disc development and injury" is well executed and clearly written.

*Please avoid overusing abbreviations, I suggest spelling out TMJ, EC, MP, MC etc.

*Please indicate how many animals or cells from individual animals were used in each set of data. It is hard at this point to evaluate rigour.

REVIEWER COMMENTS

Reviewer #1 (Remarks to the Author):

This study from the Zhu lab aims to delineate a comprehensive roadmap of TMJ disc cells at various post-natal ages in mice. The study team uses single-cell RNAseq to identify 4 cell types, including fibroblasts, endothelial cells, macrophages and mural cells, with 9 distinct clusters. Most notably, NOTCH3+/THY1+ MCs were identified to be progenitor cells that self-renew and have multi-potent ability both in vitro and in vivo. The authors use a TMJ disc injury model and lineage tracing to argue that MCs differentiate into other disc cell types during repair.

Overall this is an important study that addresses a critical medical problem. TMJ biology is severely understudied and treatment for TMDs are limited. The sequencing data provides key information that would advance the understanding of heterogenous TMJ disc cell types and thus represents a significant advance to the field. The computational analyses are comprehensive. However there are several weaknesses that need to be addressed.

The team uncovered 4 cell types, whereas 2 cell types were previously historically known as fibroblasts and chondrocyte-like cells. However it is unclear whether chondrocytes were discovered in their analyses as chondrocytes are not mentioned. This is an important point to make in defining cell types and distinguishing the disc from the condyle and also for reconciling previous studies to this new scRNAseq data.

This is a very important comment. We totally agree that identification of chondrocyte-like cells in our scRNA-Seq data would be very crucial for establishing unified theoretical system of TMJ disc types from previous studies and our scRNA-Seq results.

To address this important question, we used previous published marker genes of chondrogenic differentiation or cartilage formation to identify the chondrocyte-like cells in our 4 main cell types(1-3). We found that expressions of both chondrogenic differentiation markers such as *Sox5*, *Sox6*, *Sox9* and chondrocyte metabolism markers such as *Chad*, *Comp*, *Col11a1* were enriched in FBs cluster (Figure 2e). When we further compared expressions of these chondrocyte related markers between FB subclusters, we found that expressions of these genes were mainly enriched in FB5 and FB6 (Figure 2f). Guided by GO analyses, FB5 and FB6 were also found closely related to chondrocyte differentiation and ossification (Figure S2b). Therefore, we defined these 2 FB clusters as chondrogenesis related fibroblast 1 and 2, and other 4 FB clusters as non-chondrogenic fibroblasts, with additional definition according to their DEGs (Figure S4a).

To further verify our findings by bioinformatic analyses, we singled out *Chad* as the DEG of chondrogenesis related FBs, and *Sfrp2* as the DEG of non-chondrogenic FBs, for further cellular localization of different clusters (Figure 2j). RNA FISH showed that *Sfrp2*⁺/*Chad*⁻ non-chondrogenic fibroblasts were located at the border of anterior and posterior bands of the disc, near the attachment, while *Sfrp2*⁻/*Chad*⁺ chondrocyte-like disc cells were mainly located in the intermediate zone, which finding was consistent with previous reports by transmission electron microscopy(4). At the same time, we found that the proportion of *Sfrp2*⁺ fibroblasts in the articular disc was gradually decreased during postnatal growth and aging, while the proportion of *Chad*⁺ fibroblasts showed opposite trends (Figure 2k-m).

The majority 88% of the disc population are fibroblasts. This population is highly studied in regenerative medicine/tissue engineering. While a section of the study describes the 6 FB clusters, the authors missed the opportunity to clearly define and name those FB populations, their markers, and describe their

speculated functions. Furthermore, since multiple time points are collected authors should expand on how they propose each FB population contributes to disc development, maintenance, and possible age-dependent disease. A proposed model and/or table of the different FB populations at varying timepoints and their putative functions would be highly valuable to the field.

According to your valuable suggestion, we re-defined the 6 FBs subclusters by their DEGs and GO analyses. We defined FB5 and FB6 as chondrogenesis related fibroblast 1 and 2, based on their enriched expressions of the chondrogenesis and chondrocyte homeostasis related genes (Figure 2f), as well as the GO analyses showing FB5 and FB6 were closely related to chondrocyte differentiation and ossification (Figure S2b). In addition, we defined other 4 FB clusters as non-chondrogenic fibroblasts, with additional definition according to their DEGs (Figure S4a). For instance, FB2 was defined as proliferation related non-chondrogenic FB cluster for its high expression of *Mki67* and *Top2a*(5, 6), FB3 was defined as apoptosis related non-chondrogenic FB cluster for increased percentage of mitochondria(7), and FB4 was defined as chondrogenesis inhibition related FB cluster for specific expression of *Tspan15*, *Procr* and *Rspo2*(3) (Figure 2g-i).

To further clarify diverse roles of FB clusters at different postnatal stages, we chose our 3d scRNA-seq data, which is the most active stage for cell development and differentiation, and used CytoTRACE (v.0.3.3) to predict differentiation states of different FB clusters at this stage(8). We found that the non-chondrogenic fibroblast 2 (proliferation) cluster showed the highest CytoTRACE value, which was estimated with optimum developmental potential (8). On the other hand, we also found the chondrogenesis related FB1 and FB2 were at the relatively later order of CytoTRACE value (Figure S4a-e). These findings enlighten us again that the chondrocyte-like FBs in TMJ disc are

terminally differentiated cells with limited capacity of self-repair and being associated with aging and injury(9, 10).

The description of mural cells as both a niche and as progenitor cells is a bit confusing. Do the authors mean perivascular niche? If the authors argue MCs are niche cells, which cells do they provide a niche for? Please clarify.

The reason of describing MCs as niche cells is because we found that MCs have active signaling crosstalk with other cell types. We totally agree this speculation lacks solid evidence and made our description of MCs functions confusing. Therefore, we deleted the description of 'cell niche provided by MCs'. Thank you for this thoughtful comment.

The title reads a bit awkward. It's unclear if the authors are referring to the perivascular location as the niche or the NOTCH3+/THY1+ MCs themselves. NOTCH3+/THY1+ MCs as both a niche cell and progenitor cell is confusing. MC/progenitor cell function during disc development was not directly validated in this study, which would require loss of function studies. Unless authors can validate developmental function during development, it's recommended to remove from the title.

Followed your suggestion, the title has been changed as follows: Single-cell transcriptional atlas revealed that the resident progenitor cells function in mouse temporomandibular joint disc injury.

In Figure 3c and 3f, immunostainings for PCAM1 and C1QA are shown separately and not co-localized as described in results section line 552. TMJ

disc tissue is quite thin and cell constituents can vary dependent on different sections. Its difficult to conclude ECs and Macrophages work together in the anterior band without showing the two populations in the same section. It seems that PCAM1 is localized solely in the disc lining the superior joint cavity, whereas C1QA is found in both super/inferior joint cavity. How does this specific location correlate to their proposed function and relationship to each other?

Our speculation that 'The colocalization of C1QA⁺ MPhs and PECAM-1⁺ ECs in this area suggests that a close interaction between ECs and MPhs may contribute to the modulation of vascular function and the inflammatory response' is based on the relatively closed spatial relationship between the two cell types in the anterior band, as well as previous reports considering signal interactions between MPhs and ECs may participate in immune response tissue injury(11, 12). However, we totally agree with your concern that our current evidence is very limited to support this speculation, a series of experiments and much more comprehensive studies are needed to clarify the signal and functional connection between these two cell types. Since our current study is focusing on the functional role of mural cells as progenitors in the disc, we deleted the confusing sentence and leave this speculation to be elucidated in future work.

Figure 4 is very interesting. However, what type of "functional disc cells" do MCs become? Please define "functional cell". This goes back to the question of what is a disc FB?

'Functional cells' was used to define cells that participate in disc tissue repair, which cells were supposed to be part of FBs. To verify this speculation, we performed combined FISH and antibody staining to co-localize different FBs clusters and *Myh11*⁺ MCs lineage. Interestingly, we found that when *Myh11*⁺

MCs lineage migrated toward disc injury site, considerable number of MCs lineage started to express *Sfrp2* (RFP⁺/ *Sfrp2*⁺), but much fewer RFP⁺/*Chad*⁺ cells were observed (Figure 8l). In addition to our other experiments that showed RFP⁺ MCs lineage loss its original specific expressions of NOTCH3, THY1 and *Myh11* during disc injury repair (Figure 8a-k, S9a-c), we conclude that MCs were able to differentiate toward fibroblasts for mouse TMJ disc injury repair, specifically transformed toward non-chondrogenic fibroblasts instead of chondrogenesis related cells.

In Figure 5i-5K. The histology and pentachrome/chemical staining for bone and cartilage is not very convincing. In 5i Thy⁺ pellets seems highly variable in expressing Acan and pellet staining is unconvincing. In Fig 5K THY⁺ cells, the cartilage area (presumably blue color) appears to be missing and the lowest panel that shows “bone” looks more like fibrous tissue. It seems likely that THY⁺ cells have an inclination toward bone but not cartilage. More markers and/or better sections need to be shown to support the conclusions that THY⁺ are multipotent. To this end, it is unclear how quantification in Fig. 5K of the bone/cartilage/stroma was performed with the unconvincing histology. Please

The strategy of Movat pentachrome staining is based on previous studies (13-15). Previous studies define collagen⁺ area with yellow color as bone tissue, mucin⁺ area with blue color as cartilage tissue and fibronectin⁺ area with red color as stroma/marrow. To be more cautious about our occlusion, we switched the definition of ‘bone’, ‘cartilage’ and ‘stroma/marrow’ to ‘collagen’, ‘mucin’ and ‘fibronectin’.

We have added immunostaining of SOX9 and Aggrecan on the THY1⁺/ THY1⁻ pellets (Figure6 i-j). Consistent with the pentachrome staining results, THY1⁺ disc cells showed a significantly stronger cartilaginous capacity than THY1⁻ disc cells under chondrogenic transduction, manifesting as higher expression of

Aggrecan and Sox9. In the *ex vivo* transplantation of THY1⁺ MCs under the renal capsules of NOD-SCID mice, four-week grafts of THY1⁺ MC populations formed more collagen (yellow), fibronectin (red) and mucin (blue), however, there was no mature chondrocytes found in the renal capsule ossicles. Therefore, we speculate that it is hard for MCs to form mature cartilage without chondrogenesis induction, suggests a limited chondrogenesis capacity of MCs for disc fibrocartilage repair.

The Parabiosis model is not sufficiently described in the written results section.

Followed your suggestion, we described the parabiosis model as follows:

To rule out the possibility of a circulating source of disc tissue repair, we used a parabiosis model that generated circulation between GFP⁺ mice and TMJ injured non-GFP mice (Figure S7a). We found there was no circulating GFP⁺ cells participating in joint disc repair (Figure S7b-c)

The authors should consider using more recent nomenclature and change the term “mesenchymal progenitors” to “skeletal progenitors” to align with current skeletal stem/progenitor cell biology.

According to your advice, we have changed the ‘mesenchymal progenitors’ to ‘skeletal progenitors’.

In Fig 8: The Myh11-CreER mouse model needs to be validated in the TMJ disc by In situ or IHC for Mhy11. Please add pulse/chase timeline to figure since results and interpretation are highly dependent on recombination timing. The disc injury model is new and seems very technical for a small animal model. Its not clear how reproducibility was controlled. Please clarify surgical standardization methods. Moreover, the injury in Figure 8g-8i seems very small and its unclear how the location of the injury was determined via histology vs

tissue artifact. The injury is in the middle and distant from the anterior region. Why was this location chosen over anterior region where most of the MCs/progenitor cells were located? Were the authors testing secreted factors to encourage migration/differentiation to injury site? Please choose another pseudo-color for NOTCH given yellow also denotes overlap of green/red signal.

Followed your suggestion, we first performed combined FISH of *Myh11* and antibody staining of *Myh11-CreER* driven RFP in TMJ disc. 10 days after the first tamoxifen injection, the *Myh11-CreER* driven RFP expression has high ratio of overlap expression of RFP and *Myh11* labeling ((85.4±5.4) %) (Figure S9a-b). Thus, we suppose the Cre-driven RFP⁺ cells could effectively label *Myh11*⁺ cells under our experimental condition.

For the surgical standardization of TMJ injury model generation, we added some details in the Material and Methods session:

A vertical incision of less than 1 cm was made around the end of the zygomatic arch. Blunt tools were used to resect the muscles to expose the TMJ articular capsule. At a distance of 0.25 mm from the edge of the anterior bands of the TMJ disc, a 31G needle was used to puncture the anterior bands three times at each side, with 0.3 mm spacing between adjacent puncture holes. Muscle and skin were sutured with 6-0 Prolene sutures (DePuy Synthes, USA). Two weeks later, the animals were sacrificed for further analyses.

In our pilot study, we randomly selected histological sections from individual mice and performed HE staining, finding that the injury sites were located at similar areas in the anterior band (red rectangle), which showed that under our standardized surgical process, the TMJ disc injury model has good reproducibility.

The reason of choosing this area as puncture site (in the middle of anterior band instead of the area near attachment) is because we do not want to incidentally damage the blood vessels within the attachment during puncture, which may bring exogenous progenitors from circulation, leading to interference for our observation.

According to your advice, we have changed the pseudo-color for NOTCH from yellow to magenta. Thank you.

Fig 5 shows MCs have a propensity to make bone but how does that property relate to their function? Is bone found in the disc in disease or injury? During injury do Myh11-expressing Notch+ Thy1+ cells transition to bone cells? It seems there's a disconnect between the experiments that need some fine-tuning.

TMJ disc calcification, although was not a common disease, were still described by several previous studies(16-18). At the early stage of disc calcification, it usually manifests with asymptom and is difficult to be found, but it may lead to restricted joint function at later stage, including joint clicking, TMJ pain, and limitation of mouth opening.

The etiology of TMJ disc calcification remains unknown. Jibiki et al.(16) revealed that the incidence of disc calcifications was particularly high in cases of perforated discs and from elder human cadavers, suggesting inflammatory microenvironment, excessive mechanical stress and aging could be potential cause of the articular disc calcification.

The discussion was ok. It seems redundant of the results. More effort should be made summarize the 4 populations and their respective subpopulations and to provide speculation on their respective functions as it relates to disc tissue development, homeostasis and disease.

Thank you for the valuable comment. We have expanded our Discission, especially tried to discuss distinct functional roles of chondrogenic and non-chondrogenic FB clusters:

We also found that different FB clusters exhibit distinct DEGs that could be classified as chondrogenesis related FB clusters and non-chondrogenic FB clusters. This classification coordinates previous studies that divided disc cells into chondrocyte-like cells and fibroblasts. The chondrogenesis related FB clusters locate around intermedial zone of TMJ disc with enriched expression of chondrocyte ECM markers. These features, in addition to the latter order of CytoTRACE values, suggest these terminally differentiated chondrocyte like cells are functional for mechanical loading bearing and ECM homeostasis with limited capacity of self-repair. On the other hand, the non-chondrogenic FB clusters with prior order of CytoTRACE values, mainly locate near the anterior and posterior band attachment, and support more diverse DEG features such

as cell proliferation and chondrogenesis inhibition. These features suggest that non-chondrogenic FBs is the more active cell type in articular disc, potentially with high developmental and repair capacity.

While this study focuses on one progenitor cell population among MCs, it seems this is a very small population with not that many progeny per lineage tracing experiment. Thus there are likely other progenitor cell populations that give rise to other mature cell types that need to be discussed here. A discussion of TMJ disc progenitor cells in the context of skeletal stem/progenitor cells would be warranted here and broaden the scope of this work.

We agreed that MCs could not be the only progenitor source in the TMJ disc. In our lineage tracing experiments, we found that the *Myh11-CreER⁺* MCs appears to differentiate toward *Sfrp2⁺* non-chondrogenic fibroblasts instead of *Chad⁺* chondrogenesis related disc cells. This phenomenon, as well as our *in vitro* results showing that it is hard for MCs to form mature cartilage without chondrogenesis induction, suggests a limited chondrogenesis capacity of MCs for disc fibrocartilage repair. On the other hand, we also observed appreciable *Sfrp2⁺/ Myh11-RFP⁻* cells being activated on the anterior band. Since *Sfrp2* is critical for regulating progenitor differentiation during tissue regeneration and repairing(19-21), we consider there is probable other resident progenitor source that participate in the regulation of cellular homeostasis together with MCs in the TMJ disc.

Please discuss how this work would improve patient care/diagnoses as suggested in intro. It may also be helpful to discuss how these newly defined cell populations may shift during aging and disease.

Followed your suggestion, we have discussed the potential significance for TMD clinical diagnosis and treatment: The *Sfrp2⁺* FBs, which cells were

considered as potential functional cells for disc tissue repair, were also found mainly located in this area. This finding highlights the importance of homeostatic anterior attachment for disc function and improve the understanding of the vascular cell niche within the anterior band attachment area. More importantly, these findings suggest that this area could be a critical site for early clinical diagnosis and for specific target therapy of TMJ disc disarrangement.

Reviewer #2 (Remarks to the Author):

In this manuscript, Bi et al. performed Single-cell RNA-Seq of the TMJ disc and demonstrated that TMJ disc tissue contains 9 distinct clusters and 4 types of cells: fibroblasts, endothelial cells, macrophages and mural cells. The authors showed spatial-temporal heterogeneity of fibroblast cells throughout TMJ disc tissue during postnatal development and aging and determined the role of macrophages and endothelial cells in TMJ development. The authors also demonstrated that the Notch/Thy1 pathway are active in mural cells during TMJ postnatal development and aging and characterized mesenchymal progenitor capacities of Thy1+/Notch3+ mural cells. Overall, this is an interesting study revealing the functional role of 4 types of cells in TMJ disc tissue. However, some questions, listed below, need to be addressed.

1) For Figure 6c, the thresholds of flow cytometry should be kept in consistent. Please repeat this experiment.

In our flow cytometry experiments, we set blank group for each individual experimental group at each time point, and the threshold of each group at each timepoint is according to different blank that may has various threshold. This is for minimizing the effect from autofluorescence from different cells we prepared at each time. Followed your suggestion, we chose the plots with similar gate

threshold at each timepoint as our representative images for easier reading.

2)The authors reported that diameters of the TMJ discs were increased at different postnatal stages. Does the cell size of TMJ disc also increase at different postnatal stages?

We analyzed the diameter of disc cells at different stages using pentachromic staining sections. We found that there was no significant difference of cell diameter between different stages. However, when we analyzed cell density at different stages using ImageJ, we found that cells were significantly denser at 3d than other stages, which suggests that the increased size of TMJ disc during postnatal development were mainly attributed to the accumulation of extracellular matrix.

3)Please elaborate the relationship between distribution of different fibroblast clusters and ECM features, especially the different types of collagen. It would be interesting to detect different fibroblast clusters by IHC or IF in TMJ disc tissues.

Thank you for this thoughtful advice. Since previous studies divided TMJ disc cells into fibroblasts and chondrocyte-like cells with different ECM features, we first tried to distinguish the chondrocyte-like cells in our single cell data. We used previous published marker genes of chondrogenic differentiation or cartilage formation to identify the chondrocyte-like cells in our 4 main cell

types(1-3). We found that expressions of both chondrogenic differentiation markers such as *Sox5*, *Sox6*, *Sox9* and chondrocyte metabolism markers such as *Chad*, *Comp*, *Col11a1* were enriched in FBs cluster (Figure 2e). When we further compared expressions of these chondrocyte related markers between FB subclusters, we found that expressions of these genes were mainly enriched in FB5 and FB6 (Figure 2f). Guided by GO analyses, FB5 and FB6 were also found closely related to chondrocyte differentiation and ossification (Figure S2b). Therefore, we defined these 2 FB clusters as chondrogenesis related fibroblast 1 and 2, and other 4 FB clusters as non-chondrogenic fibroblasts, with additional definition according to their DEGs (Figure S4a).

To further verify our findings by bioinformatic analyses, we singled out *Chad* as the DEG of chondrogenesis related FBs, and *Sfrp2* as the DEG of non-chondrogenic FBs, for further cellular localization of different clusters (Figure 2j). RNA FISH showed that *Sfrp2*⁺/*Chad*⁻ non-chondrogenic fibroblasts were located at the border of anterior and posterior bands of the disc, near the attachment, while *Sfrp2*⁻/*Chad*⁺ chondrocyte-like disc cells were mainly located in the intermediate zone, which finding was consistent with previous reports by transmission electron microscopy(4). At the same time, we found that the proportion of *Sfrp2*⁺ fibroblasts in the articular disc was gradually decreased during postnatal growth and aging, while the proportion of *Chad*⁺ fibroblasts showed opposite trends (Figure 2k-m).

4)Figure 3c and 3d showed that macrophages were gradually decreased comparing the newborn mice to the aged mice. Have the authors analyzed macrophage polarization in this process? It is not clear if M1 and M2 markers are both decreased from the newborn stage to the aged stage.

This is an interesting question. Followed your question, we performed the flow cytometry for analyzing M1/M2 macrophages portion changes at different postnatal stages. According to previous studies(22-24), we chose CD86 as the

marker of M1 macrophages and CD206 as the marker of M2 macrophages. Then we used TMJ disc cells at different stages (3d, 3w, 6m, 12m) for CD86/CD206 flow cytometry (the results figure is attached below). We found that consistent with the C1QA expression changes, both CD86⁺ and CD206⁺ cell proportions were gradually decreased from 3d to 12m. 3d mice, which were at the peak of growth with adaptive immunity established, had both active M1 and M2 macrophage expressions. On the other hand, the M1/M2 ratio were kept stable within different postnatal stages, signifying the homeostasis of M1/M2 polarization in TMJ under physiological conditions.

TMJ is an active organ with complex extracellular environment, therefore, MPhs could potentially be a critical regulator for maintaining endostasis of the TMJ disc during development and injury. This speculation is awaiting more comprehensive experiments to be verified, and we hope to keep on investigating this important scientific question in our future study.

Representative flow cytometry plots of the percentage of m1/m2 macrophages in TMJ discs. (a-b) Gating strategies for the analysis of mouse TMJ disc single cells isolated from C57BL/6 mice at different stages (3 d, 3 w, 16 w, 78-82 w), FSC-A/SSC-A was used to distinguish between cell groups and exclude cell debris, and FCS-A/FSC-H was used to remove adhesion cells and screen out single cells. (c) Representative graphs of the percentages of CD86⁺ cells or CD206⁺ cells at different stages is presented (right, n = 2-3 biological replicates per group.). The proportion of CD86⁺ cells (dot on the right side of the gate) and the proportion of CD206⁺ cells (dot on the upper side of the gate) gradually decreased with age, while there was a rebound tendency at 78-82w group. (d-f) Proportion change of different types of macrophages at different stages. Each dot represents a biological duplication, one-way ANOVA followed by multiple comparisons were used for statistical analysis.

5) It has been shown that Notch signaling participates in the onset and development of TMJ-OA and inhibition of Notch signaling pathway temporally postpones the cartilage degradation of TMJ arthritis in mice (DOI: 10.1016/j.jcms.2018.04.026). In this study, the authors found that Notch signaling was activated at all postnatal stages in mural cells. Are these mice suffering from spontaneous TMJ-OA? Please discuss more about Notch signaling in TMJ-OA development.

Followed your advice, we have expanded our Discussion about NOTCH signaling in TMJ homeostasis and OA:

NOTCH signaling pathway was found with a dual role during joint cartilage metabolism, as well as being identified as a potential regulator of both catabolic and anabolic molecules in the cartilage ECM during development(25-27). Transient activation of NOTCH signaling in postnatal chondrocytes results in increased synthesis of cartilage ECM and joint maintenance, while overexpression of NOTCH signaling activates the pathway in OA cartilage(28). In TMJ, our previous study showed that NOTCH signaling was excessively activated during the onset and development of TMJOA, while partial blocking NOTCH signaling by preventing NICD release could alleviate the cartilage destruction(29). In addition, NOTCH activation was found critical for chondrogenic progenitor specifications in condylar fibrocartilage(30). These findings, in addition to our current study showing that NOTCH is active at all postnatal stages in TMJ disc, suggest potential functions of NOTCH both as the 'identity' and the regulator of MCs characteristics during TMJ cartilage development and injury repair.

Reviewer #3 (Remarks to the Author):

The paper "Single-cell transcriptional atlas revealed that the resident progenitor cell niche functions in TMJ disc development and injury" is well executed and clearly written.

Thank you very much for your praise for our work.

*Please avoid overusing abbreviations, I suggest spelling out TMJ, EC, MP, MC etc.

Followed your suggestion, we spelled out FBs, MCs, ECs, MPs as fibroblasts, mural cells epithelial cells and macrophages at the beginning of each section of the Results and Discussion.

As for TMJ, since it's a proprietary abbreviation in the craniofacial research field, we would love to keep TMJ for more succinct expression.

*Please indicate how many animals or cells from individual animals were used in each set of data. It is hard at this point to evaluate rigour.

Followed your suggestion, we have added animal numbers for each experiment in our figure legend, therefore reviewers and readers can clearly know the exact number of animals used in each set of data. Thank you.

1. Bi, R., Yin, Q., Mei, J., Chen, K., Luo, X., Fan, Y., and Zhu, S. (2020) Identification of human temporomandibular joint fibrocartilage stem cells with distinct chondrogenic capacity. *Osteoarthritis Cartilage* **28**, 842-852
2. Geng, H., Carlsen, S., Nandakumar, K. S., Holmdahl, R., Aspberg, A., Oldberg, A., and Mattsson, R. (2008) Cartilage oligomeric matrix protein deficiency promotes early onset and the chronic development of collagen-induced arthritis. *Arthritis Res Ther* **10**, R134

3. Tachibana, N., Chijimatsu, R., Okada, H., Oichi, T., Taniguchi, Y., Maenohara, Y., Miyahara, J., Ishikura, H., Iwanaga, Y., Arino, Y., Nagata, K., Nakamoto, H., Kato, S., Doi, T., Matsubayashi, Y., Oshima, Y., Terashima, A., Omata, Y., Yano, F., Maeda, S., Ikegawa, S., Seki, M., Suzuki, Y., Tanaka, S., and Saito, T. (2022) RSPO2 defines a distinct undifferentiated progenitor in the tendon/ligament and suppresses ectopic ossification. *Sci Adv* **8**, eabn2138
4. Detamore, M. S., Hegde, J. N., Wagle, R. R., Almarza, A. J., Montufar-Solis, D., Duke, P. J., and Athanasiou, K. A. (2006) Cell type and distribution in the porcine temporomandibular joint disc. *J Oral Maxillofac Surg* **64**, 243-248
5. Uxa, S., Castillo-Binder, P., Kohler, R., Stangner, K., Muller, G. A., and Engeland, K. (2021) Ki-67 gene expression. *Cell Death Differ* **28**, 3357-3370
6. Chen, L., Chou, C. L., and Knepper, M. A. (2021) Targeted Single-Cell RNA-seq Identifies Minority Cell Types of Kidney Distal Nephron. *J Am Soc Nephrol* **32**, 886-896
7. Lun, A. T. L., Riesenfeld, S., Andrews, T., Dao, T. P., Gomes, T., participants in the 1st Human Cell Atlas, J., and Marioni, J. C. (2019) EmptyDrops: distinguishing cells from empty droplets in droplet-based single-cell RNA sequencing data. *Genome Biol* **20**, 63
8. Gulati, G. S., Sikandar, S. S., Wesche, D. J., Manjunath, A., Bharadwaj, A., Berger, M. J., Ilagan, F., Kuo, A. H., Hsieh, R. W., Cai, S., Zabala, M., Scheeren, F. A., Lobo, N. A., Qian, D., Yu, F. B., Dirbas, F. M., Clarke, M. F., and Newman, A. M. (2020) Single-cell transcriptional diversity is a hallmark of developmental potential. *Science* **367**, 405-411
9. Berkovitz, B. K., and Pacy, J. (2000) Age changes in the cells of the intra-articular disc of the temporomandibular joints of rats and marmosets. *Arch Oral Biol* **45**, 987-995
10. Oberg, T., Carlsson, G. E., Fajers, C. M., and Bergman, F. (1966) Ageing of the human temporo mandibular disk with special reference to the occurrence of cartilaginous cells. *Odontol Tidskr* **74**, 122-129
11. Hernandez, G. E., and Iruela-Arispe, M. L. (2020) The many flavors of monocyte/macrophage--endothelial cell interactions. *Curr Opin Hematol* **27**, 181-189
12. Marichal, T. (2020) Endothelial cells instruct macrophages on how to Respond to lung injury. *Nat Immunol* **21**, 1317-1318
13. Ambrosi, T. H., Sinha, R., Steininger, H. M., Hoover, M. Y., Murphy, M. P., Koepke, L. S., Wang, Y., Lu, W. J., Morri, M., Neff, N. F., Weissman, I. L., Longaker, M. T., and Chan, C. K. (2021) Distinct skeletal stem cell types orchestrate long bone skeletogenesis. *Elife* **10**
14. Marecic, O., Tevlin, R., McArdle, A., Seo, E. Y., Wearda, T., Duldulao, C., Walmsley, G. G., Nguyen, A., Weissman, I. L., Chan, C. K., and Longaker, M. T. (2015) Identification and characterization of an injury-induced skeletal progenitor. *Proc Natl Acad Sci U S A* **112**, 9920-9925
15. Ambrosi, T. H., Marecic, O., McArdle, A., Sinha, R., Gulati, G. S., Tong, X., Wang, Y., Steininger, H. M., Hoover, M. Y., Koepke, L. S., Murphy, M. P., Sokol, J., Seo, E. Y., Tevlin, R., Lopez, M., Brewer, R. E., Mascharak, S., Lu, L., Ajanaku, O., Conley, S. D., Seita, J., Morri, M., Neff, N. F., Sahoo, D., Yang, F., Weissman, I. L., Longaker, M. T., and Chan, C. K. F. (2021) Aged skeletal stem cells generate an inflammatory degenerative niche. *Nature* **597**, 256-262
16. Jibiki, M., Shimoda, S., Nakagawa, Y., Kawasaki, K., Asada, K., and Ishibashi, K. (1999) Calcifications of the disc of the temporomandibular joint. *J Oral Pathol Med* **28**, 413-419

17. Han, W. H., Meng, J. H., Li, G., and Ma, X. C. (2019) Diagnosis of Bilateral Calcifications of Temporomandibular Joint Disc by Image Fusion. *The Journal of craniofacial surgery* **30**, e597-e598
18. Wang, Y. H., Li, G., Ma, R. H., Zhao, Y. P., Zhang, H., Meng, J. H., Mu, C. C., Sun, C. K., and Ma, X. C. (2021) Diagnostic efficacy of CBCT, MRI, and CBCT-MRI fused images in distinguishing articular disc calcification from loose body of temporomandibular joint. *Clin Oral Investig* **25**, 1907-1914
19. Alfaro, M. P., Pagni, M., Vincent, A., Atkinson, J., Hill, M. F., Cates, J., Davidson, J. M., Rottman, J., Lee, E., and Young, P. P. (2008) The Wnt modulator sFRP2 enhances mesenchymal stem cell engraftment, granulation tissue formation and myocardial repair. *Proc Natl Acad Sci U S A* **105**, 18366-18371
20. Yang, H., Li, G., Han, N., Zhang, X., Cao, Y., Cao, Y., and Fan, Z. (2020) Secreted frizzled-related protein 2 promotes the osteo/odontogenic differentiation and paracrine potentials of stem cells from apical papilla under inflammation and hypoxia conditions. *Cell Prolif* **53**, e12694
21. Li, G., Han, N., Yang, H., Zhang, X., Cao, Y., Cao, Y., Shi, R., Wang, S., and Fan, Z. (2020) SFRP2 promotes stem cells from apical papilla-mediated periodontal tissue regeneration in miniature pig. *J Oral Rehabil* **47 Suppl 1**, 12-18
22. Zhang, M., He, Y., Sun, X., Li, Q., Wang, W., Zhao, A., and Di, W. (2014) A high M1/M2 ratio of tumor-associated macrophages is associated with extended survival in ovarian cancer patients. *J Ovarian Res* **7**, 19
23. Yan, C., Wang, M., Sun, F., Cao, L., Jia, B., and Xia, Y. (2021) Macrophage M1/M2 ratio as a predictor of pleural thickening in patients with tuberculous pleurisy. *Infect Dis Now* **51**, 590-595
24. Oshi, M., Tokumaru, Y., Asaoka, M., Yan, L., Satyananda, V., Matsuyama, R., Matsushashi, N., Futamura, M., Ishikawa, T., Yoshida, K., Endo, I., and Takabe, K. (2020) M1 Macrophage and M1/M2 ratio defined by transcriptomic signatures resemble only part of their conventional clinical characteristics in breast cancer. *Sci Rep* **10**, 16554
25. Dong, Y., Jesse, A. M., Kohn, A., Gunnell, L. M., Honjo, T., Zuscik, M. J., O'Keefe, R. J., and Hilton, M. J. (2010) RBPjkappa-dependent Notch signaling regulates mesenchymal progenitor cell proliferation and differentiation during skeletal development. *Development* **137**, 1461-1471
26. Mead, T. J., and Yutzey, K. E. (2009) Notch pathway regulation of chondrocyte differentiation and proliferation during appendicular and axial skeleton development. *Proc Natl Acad Sci U S A* **106**, 14420-14425
27. Kohn, A., Dong, Y., Mirando, A. J., Jesse, A. M., Honjo, T., Zuscik, M. J., O'Keefe, R. J., and Hilton, M. J. (2012) Cartilage-specific RBPjkappa-dependent and -independent Notch signals regulate cartilage and bone development. *Development* **139**, 1198-1212
28. Liu, Z., Chen, J., Mirando, A. J., Wang, C., Zuscik, M. J., O'Keefe, R. J., and Hilton, M. J. (2015) A dual role for NOTCH signaling in joint cartilage maintenance and osteoarthritis. *Sci Signal* **8**, ra71
29. Luo, X., Jiang, Y., Bi, R., Jiang, N., and Zhu, S. (2018) Inhibition of notch signaling pathway temporally postpones the cartilage degradation progress of temporomandibular joint arthritis in mice. *J Craniomaxillofac Surg* **46**, 1132-1138

30. Ruscitto, A., Scarpa, V., Morel, M., Pylawka, S., Shawber, C. J., and Embree, M. C. (2020) Notch Regulates Fibrocartilage Stem Cell Fate and Is Upregulated in Inflammatory TMJ Arthritis. *J Dent Res* **99**, 1174-1181

REVIEWERS' COMMENTS

Reviewer #1 (Remarks to the Author):

The authors were quite responsive to initial critiques. The manuscript has improved significantly. This is an important study that will provide the groundwork for the TMJ field in defining disc cell populations and their functions in development, homeostasis and injury.

Reviewer #2 (Remarks to the Author):

The authors have adequately addressed this reviewer's comments. The manuscript is ready to be accepted.

Reviewer #3 (Remarks to the Author):

No further questions.